# Programming conformational cooperativity to regulate allosteric protein-oligonucleotide signal transduction

Yuan Liang [1,2,8], Yunkai Qie[3,4,5,6,8], Jing Yang[2], Ranfeng Wu[1], Shuang Cui[1], Yuliang Zhao [3,4,5], Greg J. Anderson[7], Guangjun Nie [3,4,5], Suping Li [3,4,5] ✉ & Cheng Zhang [1] ✉

Conformational cooperativity is a universal molecular effect mechanism and plays a critical role in signaling pathways. However, it remains a challenge to develop artificial molecular networks regulated by conformational cooperativity, due to the difficulties in programming and controlling multiple structural interactions. Herein, we develop a cooperative strategy by programming multiple conformational signals, rather than chemical signals, to regulate protein-oligonucleotide signal transduction, taking advantage of the programmability of allosteric DNA constructs. We generate a cooperative regulation mechanism, by which increasing the loop lengths at two different structural modules induced the opposite effects manifesting as down- and up-regulation. We implement allosteric logic operations by using two different proteins. Further, in cell culture we demonstrate the feasibility of this strategy to cooperatively regulate gene expression of PLK1 to inhibit tumor cell proliferation, responding to orthogonal protein-signal stimulation. This programmable conformational cooperativity paradigm has potential applications in the related fields.

Biosystems control diverse cellular functions and signal processing through various molecular signal pathways, where signal transduction mechanisms serve as bridges to connect the complex communications between different molecules[1–8]. For example, transcription factors (TF) can cooperatively bind to gene regulatory elements to transduce signals to regulate gene expression[1]. Signal transductions have evolved in a wide range of molecular pathways and signal networks, realizing flexible intermolecular signaling between proteins and oligonucleotides, proteins and other proteins, or polysaccharides and proteins[9–15].

Conformational signals are essential in the control of signal networks[16–22], which involves aspects of the amplitude of structural variation and multiple molecular interactions[19–23]. In nature, conformational signals usually function together to precisely regulate biological signaling pathways in a cooperative manner (e.g., macromolecule folding, structural formation and macromolecular ensembles), termed conformational cooperativity[24–29]. Recent progress in structural and biochemical studies has proven that the concept of the cooperativity has expanded to describing and regulating complex biological processes, including system behaviors and regulatory

[1]School of Computer Science, Key Lab of High Confidence Software Technologies, Peking University, 100871 Beijing, China. [2]School of Control and Computer Engineering, North China Electric Power University, 102206 Beijing, China. [3]CAS Key Laboratory for Biomedical Effects of Nanomaterials & Nanosafety, CAS Center for Excellence in Nanoscience, National Center for Nanoscience and Technology, 100190 Beijing, China. [4]University of Chinese Academy of Sciences, 100049 Beijing, China. [5]GBA Research Innovation Institute for Nanotechnology, Guangzhou 510530, China. [6]Department of Urology, Tianjin Institute of Urology, The Second Hospital of Tianjin Medical University, Tianjin 300211, China. [7]QIMR Berghofer Medical Research Institute, Royal Brisbane Hospital, Herston, Queensland 4029, Australia. [8]These authors contributed equally: Yuan Liang, Yunkai Qie. ✉e-mail: lisuping@nanoctr.cn; zhangcheng369@pku.edu.cn

mechanisms[24,25,30–32]. The conformational cooperativity phenomenon provides unique and alternative signaling controls, enabling more precise and flexible regulations on signaling pathways, gene networks and bioreceptors[1,30–36]. The examples of riboswitches show that nature RNAs adopt the cooperativity to achieve modulations of gene expression[37,38]. Take an example, a cooperative tertiary motif interaction was recently found to be able to regulate xrRNA folding and the robustness of Xrn1 resistance, where the two loop length variations in an RNA pseudoknot (e.g., 2 bp to 7 bp) can precisely modulate $Mg^{2+}$ dependency[38].

Allostery, wherein ligand binding induces dynamic structural changes at a distant site, is widely present in regulation of protein activity[37–42]. Currently, there is considerable interest in constructing artificial allosteric molecular systems, where the molecular information is recognized, transmitted and regulated mainly by conformational signals, instead of chemical signals[43–45]. In particular, DNA based allosteric systems have been well developed to construct nanoprobes, biosensors and nanoswitches for applications of controlled drug-release, point-of-care diagnostics and in vivo imaging[46–49]. In addition, complex artificial molecular systems, combing ligand and oligonucleotide interactions together, have been established to perform a signal transduction function via the allosteric controls, e.g., antibody induced DNA receptors, aptamer binding platforms, nucleic acid nanoswitches[50–58]. However, in contrast to existing conformational cooperativity in nature, the cooperative regulation in artificial allosteric transduction nanosystems has been rarely explored. It remains a great challenge to develop such systems due to the difficulties in programming and controlling of the multiple molecular structure interactions, correlations and transmissibilities.

Taking inspiration from the conformational cooperativity in nature, we in this study established a cooperative allosteric signal transduction platform (CAST; Fig. 1a and Supplementary Fig. 1), where the protein-oligonucleotide signal transduction was regulated by programming cooperative conformational signals, rather than chemical signals. Different from the reported allosteric ligand-protein transduction systems that generally use single conformational signals, our system perform CAST functions by using multiple conformational signals deriving from the ligand-binding and the loop-structure, to produce a cooperative effects. Here, we generated a dual loop regulator-based CAST regulation mechanism with unique up- and down-manipulation, based on the conformational variations and interactions. Additionally, we construct logic circuits by employing modular CAST mechanism. We then implemented a series of CAST logic operations in response to two proteins: thrombin (Thr) and streptavidin (SA; Fig. 1b). The CAST strategy was also extended to antisense oligonucleotides (ASOs) based cellular gene regulation, in which thrombin and streptavidin were used to orthogonally control the expression of model gene GFP and gene PLK1 which is closely associated with tumor cell proliferation, in a orthogonal and logical manner. It is revealed that conformational cooperativity instead of chemical signal can also provide an efficient tool to control signal transduction systems with flexible programmability, fine tunability and diverse orthogonality. It exhibits potential applications in biomolecule detection, genetic engineering and molecular signaling.

## Results
### Design of an allosteric protein-oligonucleotide signal transduction system

The concept of a basic allosteric signal transduction (AST) is illustrated in Fig. 1a, where the input protein specifically binds to a receptor, resulting in a conformational change to trigger a DNA output at a remote site. It is interesting to note that the input protein and output DNA signals are indirectly connected by the allostery, which greatly increases the signaling flexibility and orthogonality. We here describe an allosteric protein-oligonucleotide signal transducer that takes a

protease thrombin as input, and causes DNA conformational changes to produce an output DNA D. In the design, a DNA complex, CDAB, serves as a basic allosteric module (Fig. 1c). DNA C is the main structural frame of the transducer, which contains five functional domains: (1) an 18 nt binding domain that preferentially hybridizes with strand D; (2) a self-complementary region to facilitate the formation of an intramolecular hairpin structure; (3) and (4) adjustable loop spacer regions; (5) hybridization sites at both ends to bind with DNA A and B.

We first tested AST system using a thrombin triggered aptamer-binding method. To implement the allosteric transduction, the target aptamer sequences were attached at the two ends of DNA A and B to facilitate the thrombin binding (Fig. 1c). Initially, strand C preferentially hybridizes with strand D to form a metastable hybridization state. The binding of thrombin to the aptamers induces a conformational change that brings domains (1) and (2) close to each other to form an intramolecular hairpin structure, thus releasing DNA D (Supplementary Figs. 2 and 3).

We used polyacrylamide gel electrophoresis (PAGE) and fluorometry assays to investigate the behavior of AST (Fig. 1d–g, and Supplementary Figs. 4–6). In the presence of thrombin, a gel band was newly generated corresponding to released DNA D, confirming the successful allosteric operation (Fig. 1d, lanes 4 and 5). We then used a fluorescent probe-quencher system to demonstrate the AST activity, where the fluorophore, FAM, and quencher, BHQ1 (on DNA D), were in close proximity at the initial state. Upon thrombin binding to the DNA complex, DNA D is released from the hairpin C (Fig. 1e). As expected, a significant fluorescent signal increase was obtained in presence of thrombin (Fig. 1f, red line), while no fluorescence increase was observed in the absence of thrombin (Fig. 1f, black line). In addition, the fluorescence was dependent on the concentration ratio between the DNA complex and thrombin, confirming that the signal transduction was indeed induced by thrombin binding (Fig. 1g). Moreover, we also developed two other basic allosteric transduction modules triggered by streptavidin or PDGF-BB (Fig. 1h, and Supplementary Figs. 7–15). Interestingly, the varying concentrations of protein inputs elicit distinct AST effects. Within the range of concentration ratios of DNA complex/protein from 1:2.5 to 1:7.5, gradual decreases in fluorescence intensity were observed when the saturated thrombin concentrations were used (Supplementary Fig. 16), while such results were not found in the streptavidin-triggered system under similar ratios of streptavidin. This may be due to competition induced re-conjugation between thrombin and the DNA aptamer, causing an oversaturation that interferes with the AST effects[56].

### Cooperative regulation of the allosteric signal transduction systems

To mimic the cooperative conformation-controlled signal pathways found in nature, we explored the possibility of regulating the allosteric transduction in a cooperative manner. We designed two loop regulators T1 and T2 at different structural modules (Fig. 2a) based on the hypothesis that the two different structural domains could simultaneously affect the molecular interactions to perform CAST. We first validated the CAST regulation by varying length combinations of regulators T1 and T2 under a gradient of thrombin concentrations. Here, the loop length of regulator T1 was set to 4, 8 or 12 nt, while T2 was 3, 16 or 25 nt (Fig. 2b and Supplementary Figs. 17 and 18). In the condition of T1 and T2 lengths of 3 nt and 4 nt, respectively, we observed a narrow distribution of fluorescence variations with the increase of thrombin concentrations (Fig. 2c(I)). Maintaining T1 length at 3 nt, then increasing the length of T2 to 8 and 12 nt, the fluorescent signal distributions become markedly wider(Fig. 2c(II), (III)). It is likely that the length increase of T2 provides additional flexibility for thrombin binding, thus facilitating the allosteric signal transduction (T2 governs an up-regulation effect). Maintaining T2 length at 12 nt, and increasing the length of T1 from 3 to 25 nt, the fluorescence

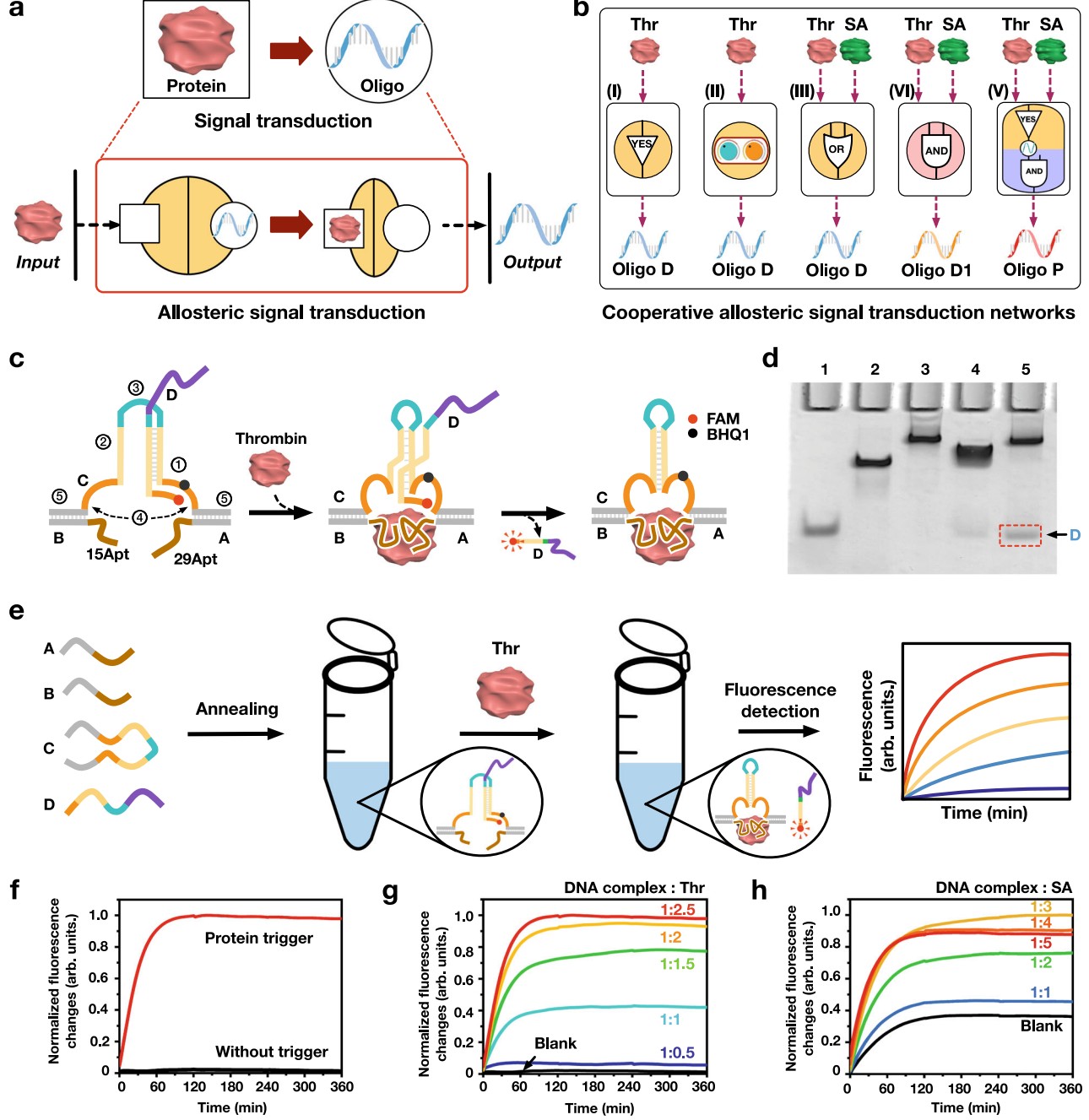

**Fig. 1 | An allosteric protein-oligonucleotide signal transduction system.**
Schematic illustrations of allosteric protein-oligonucleotide signal transduction mechanisms (**a**), cooperative allosteric signal transduction networks (**b**), respectively. Oligo D, Oligo D$_{OR}$, Oligo D$_{AND}$ and Oligo D$_{Cd}$ are the oligonucleotide outputs of the basic AST and CAST, OR logic operation, AND logic operation and cascading CAST logic operation, respectively. **c** Design of a basic allosteric signal transduction system, respectively. 15Apt and 29Apt: Two aptamers of thrombin; FAM: fluorophore; BHQ1: quencher. **d** PAGE gel analysis of the basic allosteric transduction

system. Lane 1: D; Lane 2: CAB; Lane 3: CAB + Thr; Lane 4: CDAB; Lane 5: CDAB + Thr. [DNA complex] = 0.6 μM, [Thr] = 1.5 μM. **e** The whole processes of the experimental setup in vitro. Fluorescence output of the basic allosteric transduction system triggered by thrombin (**f**) and the reactions with varying thrombin concentrations (**g**), respectively. **f**: [CDAB] = 0.6 μM, [Thr] = 1.5 μM. **g**: [CDAB] = 0.6 μM, [Thr] = 0, 0.3 μM, 0.6 μM, 0.9 μM, 1.2 μM and 1.5 μM. **h** Fluorescence output of the basic allosteric transduction system triggered by streptavidin. [C6*D3*A2B2] = 0.6 μM, [SA] = 0, 0.6 μM, 1.2 μM, 1.8 μM, 2.4 μM and 3 μM.

distribution gradually narrowed (Fig. 2c from (III) to (V)). The possible reason for this is that the length increase of T1 reduces the proximity between the two arms of the DNA hairpin, thus hindering the intrastrand displacement (T1 governs a down-regulation effect).

It is worth noting that the allosteric regulation effects were opposite for regulators positioned at different structural modules, as evidenced by the down-regulation for the length increase of T1 and

the up-regulation for the length increase of T2 (Fig. 2d). These results demonstrate that loop structures at the different modules can cooperatively control CAST with their own conformational characteristics. The largest fluorescence increase was achieved when T1 = 3 nt and T2 = 12 nt, and the lowest fluorescence intensity was obtained when T1 = 25 nt and T2 = 4 nt. Through the cooperative conformational regulation along with thrombin concentration, the

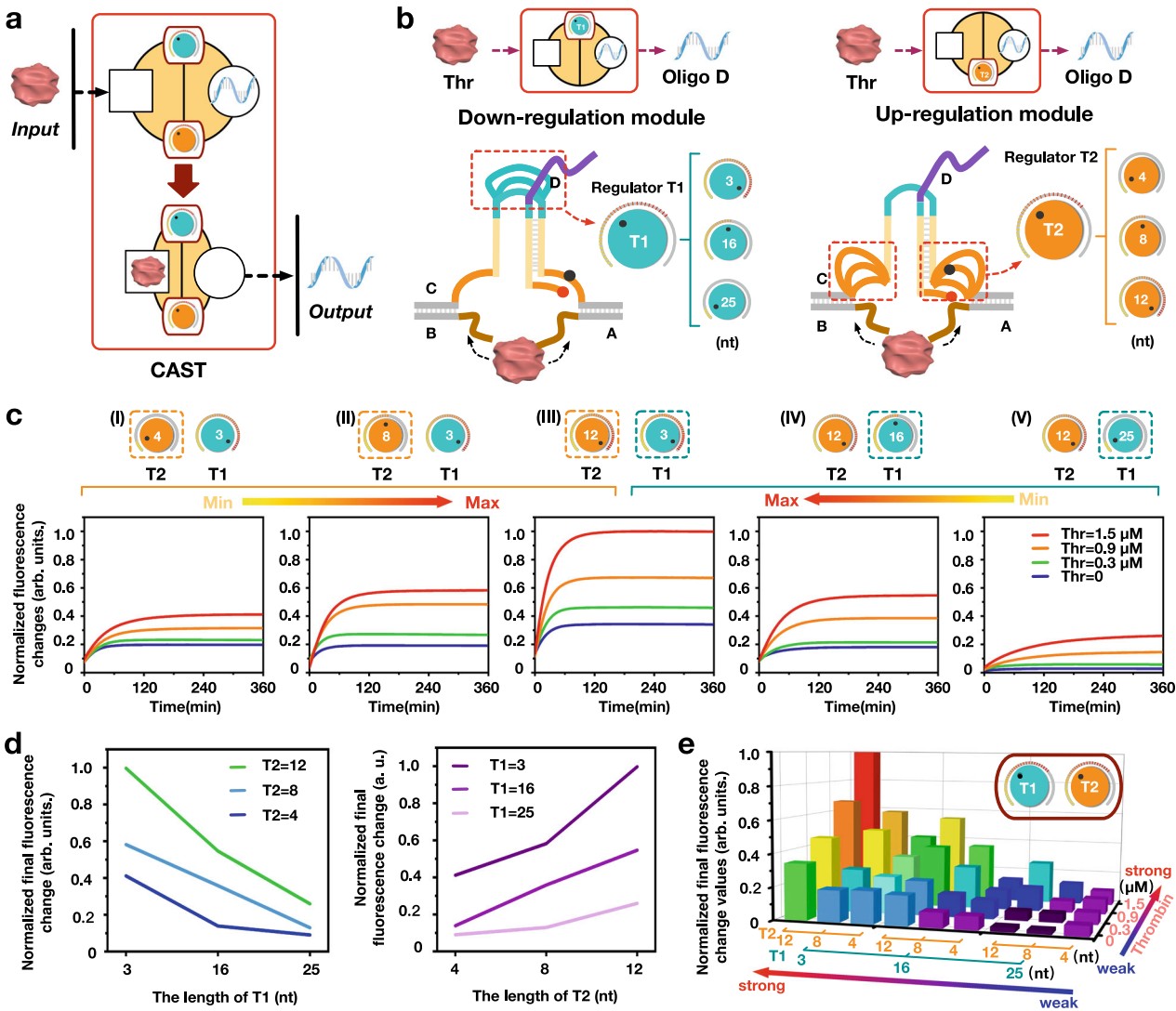

**Fig. 2 | CAST regulation by two conformational regulators, T1 and T2.**
**a** Schematic illustration of the CAST system. **b** Detailed design of regulators T1 and T2. **c** Fluorescence assay of the CAST regulation after altering the length combinations of T1 and T2 as follows (T1 length, T2 length): (I) 3 nt, 4 nt; (II) 3 nt, 8 nt; (III) 3 nt, 12 nt; (IV) 16 nt, 12 nt; (V) 25 nt, 12 nt. In each case, the concentration of DNA complex CDAB was 0.6 μM, while the concentrations of thrombin were 0, 0.3, 0.9, and 1.5 μM. **d** The down-regulation and up-regulation CAST effects of the individual regulators T1 and T2. **e** Histograms of the fluorescence data of the CAST regulation with varying T1 and T2 lengths and a range of thrombin concentrations: 0, 0.3, 0.9, and 1.5 μM.

system was capable of producing 36 CAST levels (Fig. 2e and Supplementary Fig. 19).

## Two-input allosteric logic operations based on CAST

Due to the unique orthogonal design in allosteric regulation, multiple proteins can be used as inputs to implement complex CAST with protein-oligonucleotide signals. We used two proteins, thrombin and streptavidin, to implement allosteric OR and AND gate operations as shown in Fig. 3a, e.

To construct the CAST OR gate, we designed receptor binding sites recognized by both thrombin and streptavidin (Fig. 3a and Supplementary Fig. 20). Specifically, the aptamer sequences 15Apt and 29Apt at the DNA A1 and B1 ends target thrombin, while we simultaneously modified the DNA A1 and B1 segments with two biotins for streptavidin binding. Thus, thrombin and/or streptavidin can trigger the allosteric transduction to perform the OR operations (Fig. 3b). PAGE analysis showed that when thrombin and/or streptavidin is present, new gel products representing released DNA $D_{OR}$ are generated (Fig. 3c, lanes 6, 7 and 8). In addition, fluorescence kinetics curve results also reveal that with one or both of the protein presence, the

CAST is triggered to produce significant fluorescence increases, indicating the successful fabrication of a CAST OR gate (Fig. 3d).

For a CAST AND gate, the initial structure of the DNA receptor is set as a separate state, and the corresponding allosteric transduction occurs only when both proteins (thrombin and streptavidin) are present, thus achieving the CAST regulation. In the AND gate, we designed separate protein recognition sites on the top and bottom of the DNA receptor structures, shown in Fig. 3e. Inputting thrombin or streptavidin alone is not sufficient to connect the two separate sections to perform the allosteric transduction. However, introducing both of the proteins can form the entire DNA receptor to cooperatively implement CAST and induce the release of DNA $D_{AND}$ (Fig. 3f and Supplementary Figs. 21 and 23). This is apparent in the PAGE analysis results, where inputting both thrombin and streptavidin induced a distinct product corresponding to the released DNA $D_{AND}$ (Fig. 3g). Again, we analyzed the AND operation by fluorescence assay and observed a significant signal increase only when both of the proteins were introduced (Fig. 3h). Notably, a relatively high leakage was observed when treating with only streptavidin, possibly due to the relative strong binding effect between streptavidin and biotin. In addition, more complex

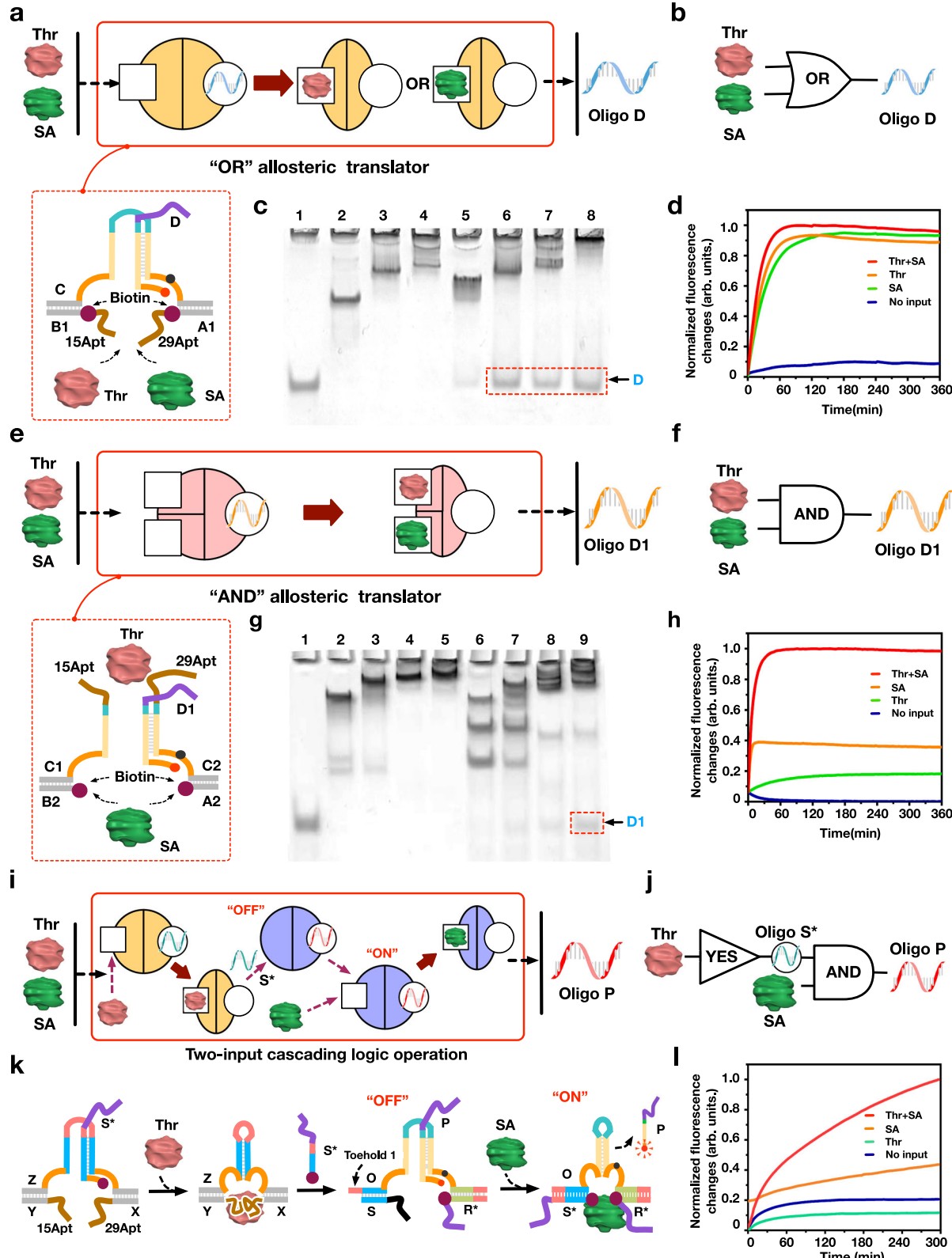

**Fig. 3 | Two-input logic operations based on the CAST strategy. a, b** Schematic illustration of an OR logic gate. **c** PAGE results of OR logic operation using DNA complex-or (CD$_{OR}$A1B1). Lane 1: D$_{OR}$; Lane 2: CA1B1; Lane 3: CA1B1 + Thr; Lane 4: CA1B1 + SA; Lane 5: complex-or; Lane 6: complex-or + Thr; Lane 7: complex-or + SA; Lane 8: complex-or + Thr + SA. **d** Fluorescence assay of OR logic operation. **e, f** Schematic illustration of AND logic gate. **g** PAGE results of AND logic operation using DNA complex-and (C1B2 + C2A2D$_{AND}$). Lane 1: D$_{AND}$; Lane 2: C1B2 + C2A2; Lane 3: C1B2 + C2A2 + Thr; Lane 4: C1B2 + C2A2 + SA; Lane 5: C1B2 + C2A2 + Thr + SA; Lane 6: complex-and; Lane 7: complex-and + Thr; Lane 8: complex-and + SA; Lane 9: complex-and + Thr + SA. **h** Fluorescence assay of AND logic operation. [DNA strands] = 0.6 μM, [Thr] = 1.5 μM, [SA] = 1.8 μM. **i** Cascading CAST circuit triggered by thrombin and streptavidin. **j** The illustrations and **k** designs of cascading CAST logic operation (using DNA complex (XI), ZS*XY and complex (XII), OD$_{Cd}$R*S), respectively. S* and D$_{Cd}$ are the upstream and downstream outputs of the cascade circuit, respectively. **l** Fluorescence results. [complex (XI)] = 0.5 μM, [complex (XII)] = 0.5 μM, [Thr] = 1.25 μM, [SA] = 1.5 μM.

cascading allosteric logic operations were implemented using both thrombin and streptavidin (Fig. 3i–l, and Supplementary Figs. 24 and 28). Overall, we demonstrate logic CAST operations using two different proteins, exploiting highly flexible orthogonality and cooperativity.

## The CAST operations with single-trigger-site

To investigate whether can be triggered by the CAST receptor with the single binding site, we constructed two kinds of CAST receptors that were designed to respond to any one of thrombin or PDGF-BB. We first designed the single-trigger-site CAST receptor using one 29 nt

aptamer to interact with protein thrombin (Fig. 4a, b, and Supplementary Fig. 29). It should be noted that 29 nt length DNA E-29apt is designed in the middle as the single binding site. Therefore, binding of thrombin to the 29 nt aptamer sequence can generate a significant conformational changes in the DNA receptor, thus inducing the close proximity of the two arms of hairpin DNA to release DNA D. In the gel results, it is clear to see that a target gel band representing the released DNA D was produced in lane 5 (Fig. 4c and Supplementary Fig. 30a–c). Additionally, a significant positive fluorescent signal was produced when triggered by thrombin. The gradual increases of fluorescent signals also can be found with gradient increasing thrombin (Fig. 4d).

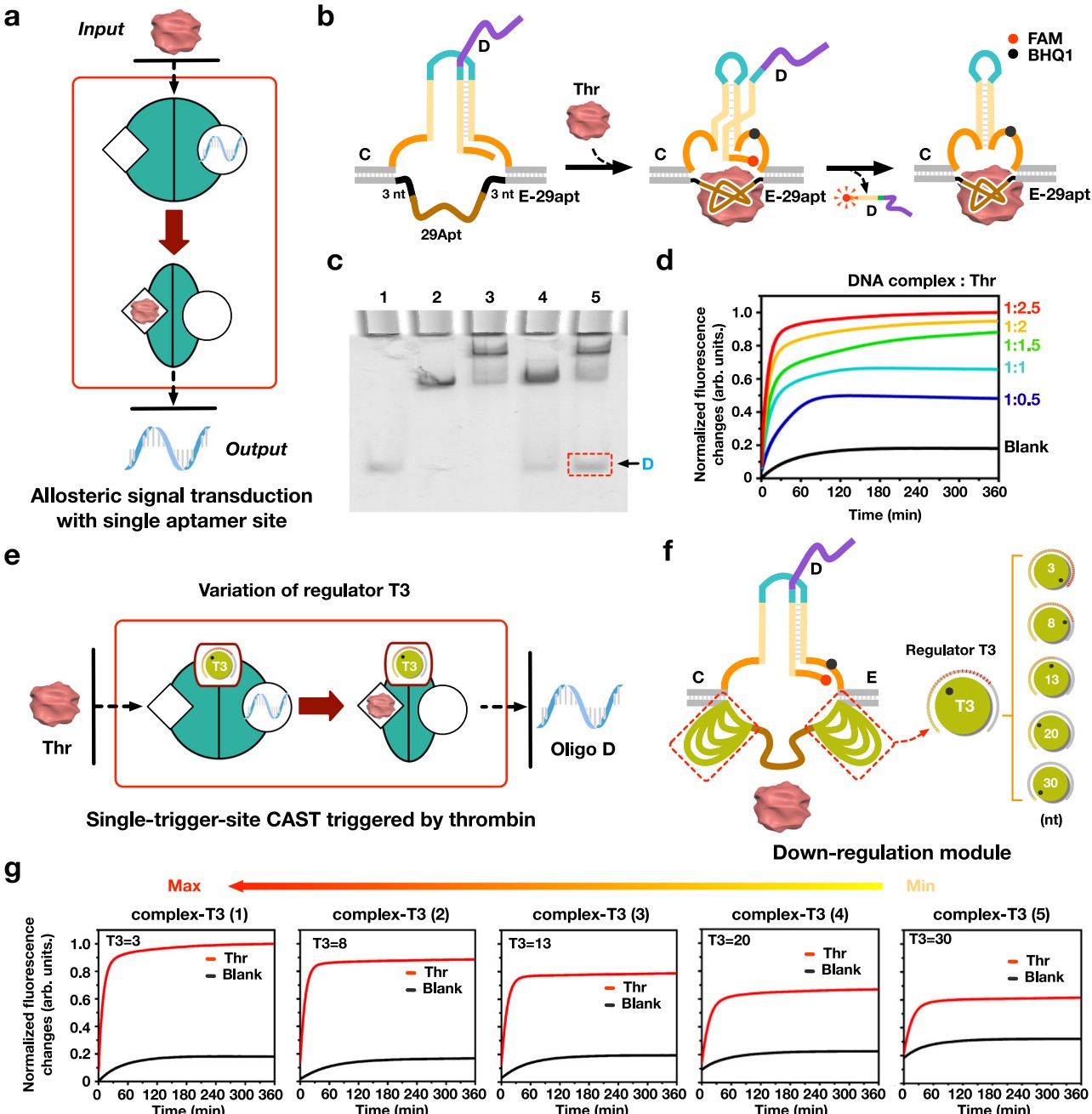

**Fig. 4 | Single-trigger-site CAST operations triggered by thrombin.**
**a**, **b** Schematic and design illustration of single-trigger-site CAST triggered by thrombin. **c** PAGE gel analysis of single-trigger-site CAST. Lane 1: D; Lane 2: CE-29apt; Lane 3: CE-29apt + Thr; Lane 4: CDE-29apt; Lane 5: CDE-29apt + Thr. [DNA complex] = 0.6 μM, [Thr] = 1.5 μM. **d** Fluorescence output of single-trigger-site CAST varying thrombin concentrations. **d** [DNA complex] = 0.6 μM, [Thr] = 0, 0.3 μM, 0.6 μM, 0.9 μM, 1.2 μM, and 1.5 μM. Illustrations (**e**) and designs (**f**) of the single-trigger-site CAST triggered by thrombin with different T3 lengths, respectively. **g** Fluorescence results of the single-trigger-site CAST triggered by thrombin with different T3 lengths of 3, 8, 13, 20 and 30 nt for complex-T3 (1), (2), (3), (4), (5). [complex-T3] = 0.6 μM and [Thr] = 1.5 μM.

Over all, our results demonstrated the well performance of the single-trigger-site CAST receptor triggered by thrombin protein.

Next, we also tested the single-trigger-site based down-regulation module by introducing loop regulator T3 with the lengths varying from 3 to 30 nt (Fig. 4e, f). Fluorescent assay was implemented, and the responding results showed the down-regulations where the gradual decreasing fluorescent signal densities were generated with the loop lengths increasing (Fig. 4g and Supplementary Fig. 30d). Meanwhile, in the gel results, the gradual decreasing band densities of released DNA D can be observed with the increase of loop lengths (Supplementary Fig. 30e, f). The experimental results demonstrated that the precise regulations also can be implemented in the single-trigger-site CAST module. We also tried to construct another single-trigger-site initiated CAST receptor by using protein PDGF-BB (Supplementary Figs. 31–33).

## Using CAST to control antisense oligonucleotide-based genetic regulation

Antisense oligonucleotides are 13–30 base pair-length oligonucleotides that can bind to target mRNAs through the complementary base pairing to block the corresponding protein translation (genetic knockdown)[59]. Recently, ASOs have been widely developed as a therapeutic genetic engineering strategy for many diseases[60–62]. With a growing number of ASOs based therapeutics, many advanced optimizations have been made to improve the delivery efficiency and the target engagement[59,61]. These improved ASOs usually adopted the strategies of specific backbone modifications thus conferring enhanced properties[62]. Meanwhile, a precise and controlled ASOs delivery mechanism is necessary to achieve safe and more effective gene-targeted therapies. We then tried to develop a CAST method to precisely control ASOs-based gene regulation based on protein triggered allosteric transduction.

We first used an antisense oligonucleotide that targets model gene GFP mRNA (ASO $D_{GFP}$), to down-regulate green fluorescent protein (GFP) expression in HeLa cells to verify the allosteric transduction gene regulation (Fig. 5a, b and Supplementary Fig. 34). We used a 2 h thrombin treatment as the protein input to trigger the release of ASO $D_{GFP}$ at room temperature (DNA Complex-G, C3$D_{GFP}$AB). We then incubated the HeLa cells with the reaction solutions containing the released $D_{GFP}$ for 24 h. The released $D_{GFP}$ was observed in the cells with both flow cytometry and confocal microscopy assays (Supplementary Figs. 35–37). In contrast, a significant loss of the fluorescent GFP signal was observed in the thrombin and free $D_{GFP}$ groups (Fig. 5c). The numbers of GFP-positive cells and mean fluorescence intensities significantly also decreased when treated with thrombin alone (Supplementary Fig. 38).

Next, we used the two conformational regulators, T1 and T2, to coordinately implement CAST regulation of GFP expression (Fig. 5d, Supplementary Figs. 39 and 40). Five varied combinations of T1 and T2 were used as follows: complex (i) C3-1$D_{GFP}$AB (16 nt, 4 nt), (ii) C3-2$D_{GFP}$AB (16 nt, 8 nt), (iii) C3-3$D_{GFP}$AB (16 nt, 12 nt), (iv) C3-4$D_{GFP}$AB (3 nt, 8 nt) and (v) C3$D_{GFP}$AB (3 nt, 12 nt; Fig. 5e). By confocal microscopy imaging, we found that complex (v), with the shortest T1 length and the longest T2 length, elicited the strongest gene down-regulation, while complex (i), with the longer T1 length and shorter T2 length, only produced a slight reduction in GFP fluorescence (Fig. 5e). Accordingly, the numbers of GFP-positive cells and their average fluorescence intensities showed the corresponding changes when regulating the release efficiency of the ASOs by the CAST scheme (Fig. 5f, g). These results suggest a potential application of CAST system to precisely control ASO-based gene regulation.

## Logic CAST regulation of functional gene expression to suppress tumor cell proliferation

We further developed a CAST platform equipped with a PLK1-specific ASO to logically regulate PLK1 gene expression which is highly associated with tumor cell proliferation, to verify CAST effectiveness (Fig. 6a, Supplementary Figs. 41 and 42). PLK1 is a highly conserved serine/threonine protein kinase that promotes malignant cell proliferation. ASO-based PLK1 gene engineering has been widely explored as an attractive method in cancer therapy[63,64].

We used both thrombin and streptavidin as inputs to implement OR and AND gates to control PLK1 ASO-based gene regulation in HeLa cells (Supplementary Figs. 43–45). In the OR gate operation, inputting thrombin and/or streptavidin to DNA Complex-OR (C4$D_{PLK1}$A1B1) resulted in significant reduction in PLK1 protein expression (Fig. 6b) and mRNA levels (Fig. 6c). Additionally, the densities of cell clones were significantly reduced in the presence of either or both input proteins (Fig. 6d, e), indicating effective inhibitory effects in HeLa cell proliferation. Specifically, the relative ratios of average cell clone numbers decreased to 55.6%, 52.8% and 50.0% in the presence of thrombin, streptavidin or both proteins, respectively, when compared with the negative control group. In this system, the inhibitory effects on cell proliferation were almost identical when using either the logic CAST regulation or direct treatment with free $D_{PLK1}$ (Fig. 6e). In the AND gate configuration (DNA Complex-AND, C1B2 + C2$D_{PLK1}$A2), the inhibition effects occurred only when both of proteins were inputted (Fig. 6f-i). The relative ratios of the average cell clone numbers were reduced to 58.6% of the control in the presence of thrombin and streptavidin, which is markedly lower than those in the presence of thrombin alone (101.7%) or streptavidin alone (94.8%) (Fig. 6i).

## Discussion

In the current study, we describe the construction and characterization of a protein-oligonucleotide allosteric signal transduction platform that can be controlled in a programmable and cooperative fashion. Through CAST strategy, the protein signals can be orthogonally transduced to any desired oligonucleotides. Meanwhile, employing conformational cooperativity, the signal transduction is precisely regulated by adjusting two loop lengths to achieve fine-tuning of the coordination. The logic operations (OR and AND gates) of the signal transduction mechanisms were established using two proteins as the triggers. In addition, we developed the single-trigger-site initiated CAST receptor, and have made the comprehensive investigations on the CAST regulations by varying the protein concentrations and loop lengths. The CAST strategy was also applied to ASOs to use common proteins combined with the conformational cooperativity to engineer genetic regulation, of two different genetic targets, GFP and PLK1. The latter was transduced into cellular functional effects, in which the tumor cell proliferation was effectively inhibited.

Our study demonstrates that the programmable CAST strategy possesses the unique features of cooperative regulation, logic operations and intracellular compatibility. There are four main findings in our current study, including flexible orthogonal design, precise allosteric regulation, multi-signal conformation cooperativity and logic control gene expression. First, the system is a flexible allosteric signal transduction platform with orthogonal design. The signal transduction mechanisms between input proteins and output oligonucleotides are mediated by conformational changes. Since the input and output signals do not directly interact with each other, flexible and expanded manipulation can be achieved by modularly substituting the recognition and responding domains. Second, the precisely allosteric regulation of the signal transduction mechanism was achieved by programming conformational signals. This is of particular significance for that the metastable hairpin DNA complex endows the CAST approach with conformational susceptibility, thus enabling the loop lengths directed transduction regulation. For example, small length variations of regulator T2 (e.g., from 1.36 nm to 4.08 nm) can induce significant fluorescent changes from 39.9% to 100% (Fig. 2c(I), (III)). Third, the system exhibits multi-signal integration through conformational cooperativity. Our study extends the concept of

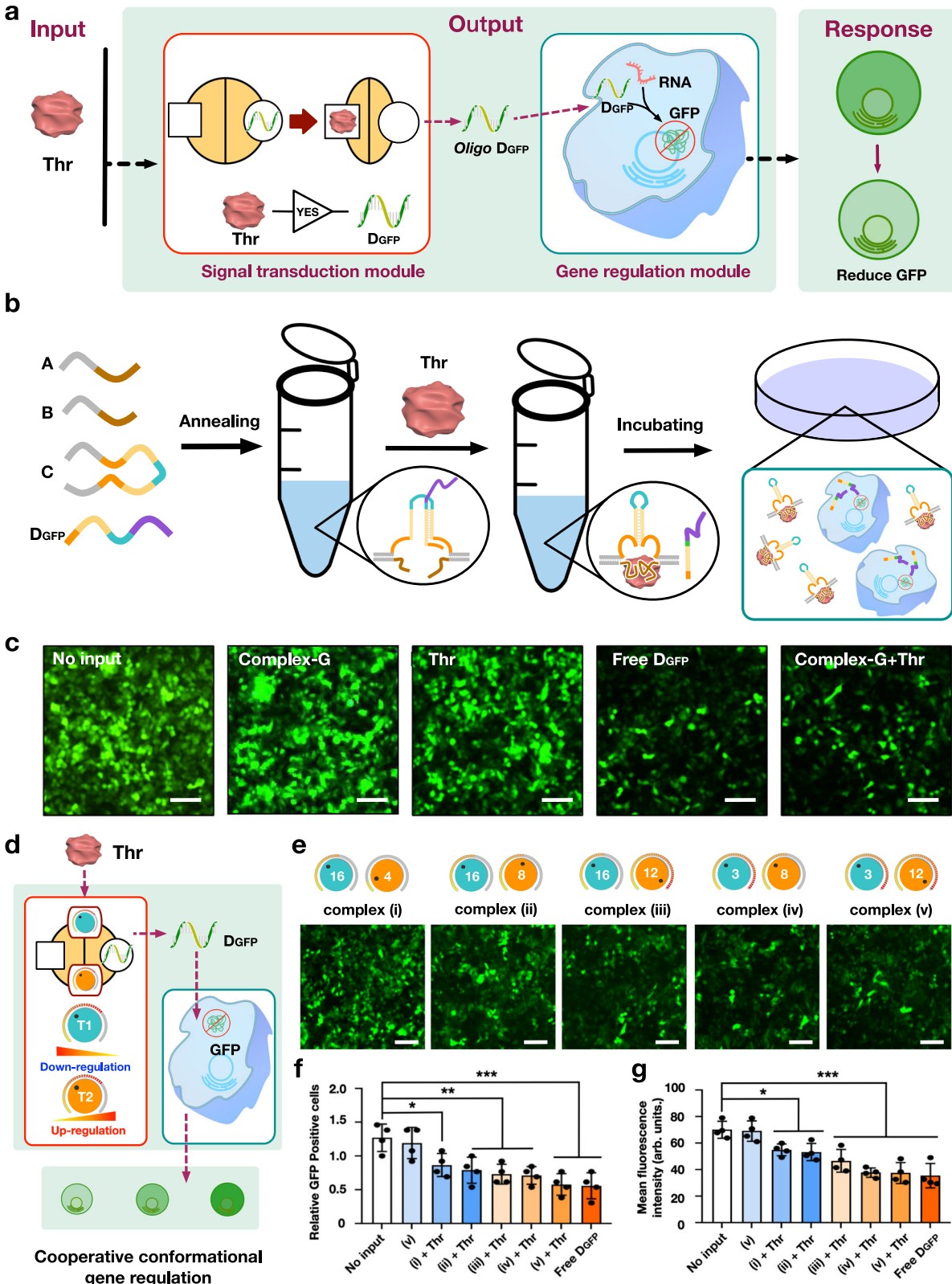

**Fig. 5 | Using CAST based ASOs to regulate cellular gene expression. a** Schematic illustration of the allosteric regulation of GFP gene expression via thrombin. **b** The whole processes of the experimental setup in vivo. **c** Confocal microscopy imaging results. Scale bars: 200 μm. Schematic illustration (**d**) and the confocal microscopy images (**e**) of CAST regulation of GFP gene expression using two regulators T1 and T2, respectively. Scale bars: 200 μm. Relative GFP-positive cells (**f**) and mean fluorescence intensities (**g**) of CAST-regulated GFP gene expression, respectively. **f**, **g** Data collected in **e** were quantified using ImageJ software and are presented as mean ± s.d. for $n = 4$ biologically independent experiments. Source data are provided as a Source Data file. Statistic analysis for **f** and **g** was performed using two-sided test (*$p \leq 0.05$, **$p \leq 0.01$, ***$p \leq 0.001$).

programmable and modular cooperativity to regulate the complex allosteric transduction behaviors, including folding formations, macromolecular ensembles and binding-induced allostery. In particular, the increases of the loop lengths in two structure modules induced the opposite CAST effects manifesting as down- and up-regulation by regulators T1 and T2, respectively. Finally, we realize the cooperative allosteric regulation of ASOs based gene expression. It is of great interest to further expand our CAST platform to control gene expression, thus enabling the use of a wider range of common proteins as orthogonal gene regulators.

Since our work presents a unique cooperative conformation approach, it provides an alternative allosteric signal regulation tool that the signal transductions and gene expressions can be precisely regulated by simply varying the structures of the DNA constructs, instead of the traditional concentration dependent methods. Therefore, the CAST platform has promising potential applications in the development of future molecular signaling systems to control gene regulation, biosensing, biocomputing, and the treatment of specific diseases (Supplementary Fig. 46). For example, the ligand-bound CAST antisense oligonucleotide technology can be established to regulate target genes for advanced intelligent diagnosis and therapy, in response to the protein triggers in cellular and complex microenvironmental conditions. On the other hand, the reported systems still have several potential limitations. For example, the operations of CAST process must depend on the specific recognition sites, which makes it difficult to work with arbitrary protein inputs, especially for the proteins without aptamer sequence. In addition, it is challenging to establish large scale and hierarchical signal transduction cascades, due to the difficulties in the designs of complex ligand binding interactions and the controls of unavoidable system leakage. Furthermore, our CAST trigger reactions currently occur in the test tubes rather than extracellular environment. Therefore, there are still some issues associated with the performances induced by extracellular or intracellular triggering. Due to the complexities of the extracellular environments, the following factors should be considered including enzymatic degradation, pH sensitivity, and the efficient delivery of the DNA strands into cells[59,65]. To overcome these hurdles, some related technologies can be introduced into CAST systems. One possible approach involves using chemical modification groups to protect the DNA receptors and released ASOs (e.g., phosphorylation by nucleic acid strands), to prevent ribozyme degradation[61,62]. Additionally, the carrier based methods also can be introduced to improve the ASOs delivery efficiency and avoid the enzymatic degradations, e.g., liposomal, nanoparticles, polymers[61,63,65]. In general, our CAST method can be improved in many aspects in future researches to develop a more versatile and practical gene regulation platform (Supplementary Fig. 47).

In summary, we have demonstrated that the programmable conformational cooperative strategy can be used to implement flexible allosteric signal transduction from proteins to oligonucleotides. We propose that this paradigm will be an efficient molecular signaling platform that allows the precisely conformational regulation of signal transduction and the flexibly programmable construct of signal pathways. The CAST platform represents an avenue to develop future molecular systems and nanomachines for potential applications of variety of biological signal detection and regulation in the related fields.

## Methods

### Materials
All oligonucleotides were purchased from Sangon Company (China). The unmodified DNA strands were purified by ULTRAPAGE. The modified DNA molecules were purified by high-performance liquid chromatography. The DNA strands were diluted with sterile ultrapure water, quantified using a Nanodrop device and used as stock solutions.

The DNA strands were stored in a refrigerator at −20 °C for ease of use. DNA sequences were designed and analyzed using NUPACK software (http://www.nupack.org). Thrombin was purchased from Solarbio (China). Streptavidin was purchased from Sigma (China). PDGF-BB was purchased from Genscript (China). Proteins were diluted with sterile ultrapure water and stored at −20 °C. The *human* cervical carcinoma cell line (HeLa) was purchased from the American Type Culture Collection (ATCC, FS-0252), authenticated by STR profiling and tested for mycoplasma contamination. All cells were tested for mycoplasma contamination and had no mycoplasma contamination. None of the cell lines used are classified as commonly misidentified lines. Cells were cultured in Dulbecco's Modified Eagle's Medium (DMEM; Wisent, Canada) supplemented with 10% fetal bovine serum (Gibco, USA) and 1% penicillin/streptomycin (Sigma-Aldrich). Primary antibodies against PLK1 (EPR19534(ab189139)) and GAPDH (6C5(ab8245)) were purchased from Abcam (Cambridge, UK). HRP-linked *goat* anti-*rabbit* secondary antibodies (PR30011) were purchased from Proteintech (China). Data related to fluorescence experiments and cell experiments were analyzed by Origin (OriginPro 2018C) and Prism (Prism 7, GraphPad software).

### Fluorescence experiments
All experiments were performed at room temperature in a $1 \times$ TAE/Mg$^{2+}$ buffer using a real time fluorescence PCR device (Agilent Technologies). Fluorophore FAM and quencher BHQ-1 were used to modify the 5' end of DNA D and the middle of DNA C, respectively. The FAM fluorescence signal was detected at 492 nm excitation and 518 nm emission. In a typical reaction, 30 μL of the solution was used for detection. The time dependence of the fluorescence results was normalized to the levels of the controls. The detection time interval was 3 min. The fluorescence results were obtained by averaging the values from three replicates.

### PAGE experiments
The reactions of the allosteric regulation were verified using native polyacrylamide gel electrophoresis (PAGE). During the experiments, a 12% acrylamide gel was made using $1 \times$ TAE/Mg$^{2+}$ buffer (12.5 mM MgCl$_2$). All samples were resolved using 100 V for 1.5–2 h at 4 °C. After staining the polyacrylamide gels with Stain-All (Sigma-Aldrich), the gels were then imaged using a Canon LIDE 100 scanner.

### DNA complex assembly procedures
DNA complexes were produced by mixing the corresponding DNA strands at equaimolar concentrations (e.g., 1 μM) in $1 \times$ TAE/Mg$^{2+}$ buffer. The sample was annealed using a polymerase chain reaction (PCR) thermal cycler using the following programs: for DNA complex assembly in fluorescence assay experiments, the annealing temperatures were 65 °C for 10 min, 60, 52, 45, 37, and 25 °C for 15 min, and finally a hold at 25 °C. For DNA complex assembly in PAGE experiments, the annealing temperatures were 95 °C for 5 min, 65, 50, 37, 25 °C for 15 min, and finally a hold at 25 °C.

### Cell experiment procedures
The DNA transducer was annealed at a concentration of 5 μM. After annealing, the triggering protein was added and incubated at room temperature for 2 h. Then the DNA samples were co-incubated with Hela cells (ATCC, FS-0252) (with the final DNA concentration as 500 nM), and the cell incubation time specifically changed in different experiments. In the cell uptaking experiments, the cell incubation time was 2 h. In the gene regulation experiment, the cell incubation time was about 48–72 h.

### Quantitative real-time PCR (RT-qPCR)
Total RNA was extracted using a Total RNA Purification kit (Biyuntian, China). Synthesis of cDNA was performed with 1 μg total RNA using a

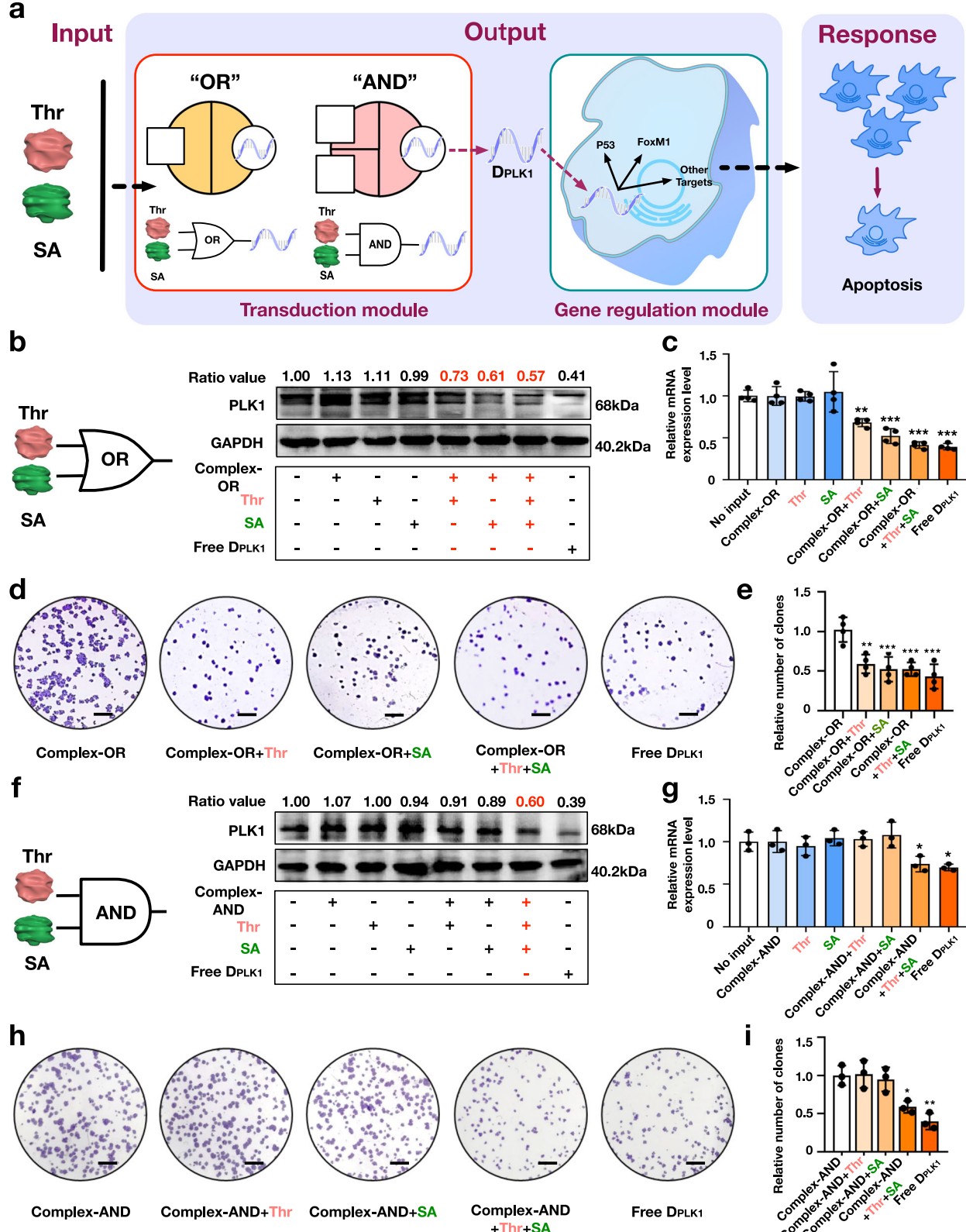

**Fig. 6 | Two-input logic CAST operations to regulate PLK1 gene expression and tumor cell proliferation. a** Schematic illustration of the logic operations. The OR logic operation effects, as assessed western blot analysis (**b**) and RT-qPCR (**c**), respectively. Cell proliferation assay results (**d**) and statistical relative clone density (**e**) of the OR logic operation, respectively. The AND logic operation results are presented similarly in panels **f**–**i**. **d**, **h** Scale bars: 2 mm. Data collected in **c** were quantified using qRT-PCR and are presented as mean ±s.d. for $n = 4$ biologically independent experiments. Data collected in **e** were quantified using ImageJ software and are presented as mean ±s.d. for $n = 4$ biologically independent experiments. Data collected in **g** were quantified using qRT-PCR and are presented as mean ±s.d. for $n = 3$ biologically independent experiments. Data collected in **i** were quantified using ImageJ software and are presented as mean ±s.d. for $n = 3$ biologically independent experimentssource data provided. Statistic analysis for **c**, **e**, **g** and **i** was performed using two-sided test (*$p \leq 0.05$, **$p \leq 0.01$, ***$p \leq 0.001$). Source data are provided as a Source Data file.

Hifair® II 1stStrand cDNA Synthesis Kit (Yeasen, China) according to manufacturer's instructions. RT-qPCR was performed using an Archimed x4 qPCR system (RocGene,China) and qPCR SYBR Green Master Mix (Yeasen, China). A standard thermal profile (95 °C, 1 min; 40 cycles at 95 °C for 30 s, 55 °C for 30 s and 72 °C for 1 min) was used for all reactions. The primer sequences used for qPCR were as follows:

PLK1-forward: 5′-GGCAACCTTTTCCTGAATGA-3′, PLK1-reverse: 5′-AATGGACCACACATCCACCT-3′; GAPDH-forward: 5′-GGAGCGAGATCC CTCCAAAAT-3′, GAPDH-reverse: 5′-GGCTGTTGTCATACTTCTCATGG-3′.

### Western blot analysis
Cells were lysed using radioimmunoprecipitation assay (RIPA) buffer (Solarbio, China) supplemented with protease inhibitor cocktail (PMSF; Solarbio, China), and the supernatant was extracted. Protein concentrations were determined using BCA (Thermo Fisher Scientific, USA). An equal amount of proteins were resolved by sodium dodecyl sulfatepolyacrylamide gel electrophoresis (SDS-PAGE; 10% acrylamide) and then transferred onto methanol-activated polyvinylidene difluoride (PVDF) membranes. After blocking in 5% BSA/TBST, the blot was incubated with the primary antibody (1:1000) at 4 °C overnight, followed by incubation with an HRP-conjugated secondary antibody (1:10,000). Finally, chemiluminescence detection was performed using ECL Prime detection reagent (GE Healthcare Life Sciences, USA) with a ChemiDoc Touch imaging system (Bio-Rad, USA).

### Construction of HeLa-GFP cell line
The HeLa cell line expressing GFP (HeLa-GFP) was constructed using the lentiviral vector GV358 (Ubi-MCS-3FLAG-SV40-EGFP-IRES), which was purchased from Genechem (China). HeLa cells were seeded in 6-well plates ($1 \times 10^6$ cells per well) and grown overnight. Next, cells were infected with GFP lentivirus according to the manufacturer's instructions, and stable clones were selected using puromycin (Solabio, China).

### Cell viability assay
HeLa cells ($1 \times 10^4$) were seeded in 96-well plates and incubated with the indicated treatments for 24–48 h. A cell counting kit-8 assay (CCK8; DOJINDO, Japan) was used for cell viability measurements. CCK-8 solution (10 μL) was added to each well containing 100 μL medium, and the samples were incubated at 37 °C for 2 h before the absorbance was measured at 450 nm.

### Colony formation assay
HeLa cells were seeded in 6-well plates at a low density (500 cells per well) and incubated for 24–48 h. Subsequently, the cells were supplemented with the selected treatments every 48 h for an additional 14 days. Colonies were fixed with paraformaldehyde (4%) for 10 min and stained with crystal violet (1%) for 15 min.

### Confocal microscopy analysis
HeLa cells were plated in 35 mm confocal microscopy dishes (glass-bottomed) and grown to approximately 60% confluency for 48 h prior to treatment. Afterwards, the cells were fixed with 4% paraformaldehyde. The nuclei were stained with DAPI (Solarbio, China), and the plasma membrane was labeled with wheat germ agglutinin (WGA; Sigma-Aldrich, USA). Confocal microscopy imaging was performed using an LSM 710 confocal microscope (Zeiss).

### Flow cytometry analysis
After seeding in 12-well plates and culturing overnight, the HeLa cells were incubated with Cy5-labeled nucleic acid strands at a final concentration of 100 nM in 500 μL DMEM complete culture medium for 3–4 h at 37 °C. After incubation, the cells were trypsinized for 3 min to obtain a suspended cells, which were then washed three times with PBS. Finally, the Cy5 fluorescence was detected by flow cytometry (Thermo Fisher Scientific, USA). Flow cytometry data were analyzed using the FlowJo software package version 10. At least 10,000 events were recorded during the sort for setting the sorting criteria and for post-sort analysis.Post-sort fractions had higher than 95% purify, as verified by flow cytometry analysis on the same machine used to sort the cells. H Gates was used to exclude debris and cell aggregates in FSC-A/SSC-A and FSC-A/FSC-H plots, and fluorescence quadrant gate was chosen to discriminate between "Cy5-positive" and "Cy5-negative" cells (Supplementary Fig. 37).

### Reporting summary
Further information on research design is available in the Nature Portfolio Reporting Summary linked to this article.

## Data availability
All fluorescence data in the main text of the article and Supplementary information were performed in three independent experiments, and the data are the average results of three experiments. Source data are provided with this paper.

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

## Acknowledgements

This work was supported by the National Key Research and Development Program of China (2021YFF1200103 and 2017YFE0130600 to C.Z.),

the National Natural Science Foundation of China (62273008 and 62073133 to C.Z. and J.Y.), the Prestudy project (31511090301 to C. Z. and J. Y.), Zhejiang Lab (NO. 2022RD0AB03 to C.Z.), the Beijing Distinguished Young Scientist program (JQ20037 to S.L.), the CAS Interdisciplinary Innovation Team (JCTD-2020-04 to S.L.), and CAS Project for Young Scientists in Basic Research (No. YSBR-036 to S.L.).

## Author contributions

C.Z. initiated the project and designed the experiments; Y.L., Y.Q., R.W. and S.C. designed the used DNA structures and performed the experiments. Y.L., and C.Z. analyzed the data. C.Z., Y.L. and S.L. wrote the paper. C.Z., Y.L, Y.Z., G.A., G.N. and S.L. discussed the designs and results of the experiments. All authors commented on the manuscript.

## Competing interests

The authors declare no competing interests.
