## [Peer Review File · Nature Communications]

Reviewers' Comments:

Reviewer #1:

Remarks to the Author:

The manuscript provides interesting insights into the development of a cooperative allosteric signal transduction platform for controlling gene expression. However, some parts of the manuscript need to be clarified and elaborated further for better understanding:

- 1.The introduction section needs more background information on the current state-of-the-art in the field of allosteric protein-oligonucleotide signal transduction systems, and how the authors' approach is different and unique from existing methods.
- 2.The authors should consider providing more comprehensive figures and diagrams to aid in the understanding of the experimental setup and the results obtained.
- 3.The manuscript would benefit from a more comprehensive review of the literature on allosteric signal transduction systems, conformational cooperativity, and antisense oligonucleotide-based genetic regulation.
- 4.The conclusion section could be strengthened by summarizing the main findings of the study and their potential implications for future research in the field.
- 5.The authors should provide more details on the limitations of the proposed approach, including potential sources of variability and the extent to which the system can be generalized to other applications.
- 6.It would be useful to include a comparison of the CAST system with other existing methods for regulating gene expression or signal transduction, highlighting the unique advantages and limitations of the CAST system.

Reviewer #2:

Remarks to the Author:

This work described a binding-induced strand displacement strategy to achieve signal transduction between proteins and oligonucleotides. Based on this strategy, the authors demonstrated multiple and logic operations between different proteins (thrombin and streptavidin), and further tried to use this system to regulate cellular gene expressions and tumor cell proliferation. The concept of this design seems solid, which may be possible to provide a new protein-oligonucleotide transduction platform. However, I do have some important concerns regarding the generality of this design and the cellular experiments.

Important concerns:

1. Generality of this design: The strand displacement described in this paper seems to rely on the protein recognition from at least two different binding sites. As investigated in this work, the streptavidin (can bind with four biotin molecules) and the thrombin (can bind with two different aptamers) worked well in this system. However, for the protein with one binding site or with only one available aptamer, would this design be still applicable? The authors tried to examine the performance on transduction of PDGF-BB with only one available aptamer (Supplementary Fig. 14), but these results are not clear. I suggest that comprehensive investigations on proteins with a single binding site or with a single binding aptamer should be performed to check the universality of this design on protein-oligonucleotide transduction in general.
2. Experiments in cells: The procedures on applications of this system in cellular manipulation are ambiguous. In Line 201-204, it seems that the transduction system was firstly incubated with the thrombin in test tubes to release the output strand before transfection into the cells. If so, all these cellular experiments (Fig. 4 and 5) would be meaningless in terms of gene regulation, as the transduction processes occurred outside cellular contexts instead of generation of the functional oligonucleotides in situ. The authors must clearly state how they performed these cell experiments. If the transduction could not be demonstrated to occur inside cells, the significance

of these results would not be as important as stated in the manuscript.

Other comments:

1. A related report (J. Am. Chem. Soc. 2013, 135, 7, 2443–2446) should be cited. This previous study on binding-induced strand displacement needs to be mentioned, compared and discussed.
2. Some experimental procedures are not clearly described. For instance, how were cells treated with these oligonucleotides? By transfection? In what concentrations?
3. The naming rule for output oligonucleotides is not defined. For instance, in Fig. 1b, what is the difference between Oligo D, D1 and P? These symbols are not explained in the manuscript.

Reviewer1

The manuscript provides interesting insights into the development of a cooperative allosteric signal transduction platform for controlling gene expression. However, some parts of the manuscript need to be clarified and elaborated further for better understanding:

Comment 1:

The introduction section needs more background information on the current state-of-the-art in the field of allosteric protein-oligonucleotide signal transduction systems, and how the authors' approach is different and unique from existing methods.

Our response:

We would like to thank the reviewer for these constructive suggestions. We have now accordingly added additional background information to strengthen the novelty, important significance and approach uniqueness of our current work in the Introduction section of the revised manuscript as below.

Page 3, lines 71-79: “Currently, there is considerable interest in constructing artificial allosteric molecular systems, where the molecular information is recognized, transmitted and regulated mainly by conformational signals, instead of chemical signals⁴³⁻⁴⁵. In particular, DNA based allosteric systems have been well developed to construct nanoprobe, biosensors and nanoswitches for applications of controlled drug-release, point-of-care diagnostics and in vivo imaging⁴⁶⁻⁴⁹. In addition, complex artificial molecular systems, combining ligand and oligonucleotide interactions together, have been established to perform a signal transduction function via the allosteric controls, e.g., antibody induced DNA receptors, aptamer binding platforms, nucleic acid nanoswitches⁵⁰⁻⁵⁸.”

Page 4, lines 86-89: “Different from the reported allosteric ligand-protein transduction systems that generally use single conformational signals, our system performs CAST functions by using multiple conformational signals deriving from the ligand-binding and the loop-structure, to produce a cooperative effect.”

Page 4, lines 91-92 : “Additionally, we construct logic circuits by employing modular CAST mechanism.”

Comment 2:

The authors should consider providing more comprehensive figures and diagrams to aid in the understanding of the experimental setup and the results obtained.

Our response:

We sincerely appreciate the reviewer for providing the thoughtful suggestions. We have now added additional pictures to better illuminate the processes of the experimental setup and the results obtained, as presented in the revised Supplementary Fig. 46 in the revised supplementary information. The corresponding descriptions were also added into the revised supplementary text as below.

For the *in vitro* experiments, short DNA strands with equal concentrations were mixed and annealed at 65°C (when modified DNA was used in the experiments, otherwise annealing was performed at 95°C). The protein with corresponding concentrations was then added and incubated for 2 hours before performing PAGE and fluorescence experiments. For the cellular experiments, the annealed DNA transducers were treated with protein triggers (e.g., thrombin and streptavidin) to release single-stranded oligonucleotides outputs. Subsequently, the DNA/protein samples from each experimental group were co-incubated with HeLa cells for certain periods. The experimental results were monitored by multiple instruments including flow cytometry, confocal microscopy, and western blot analysis, et al.

Revised Supplementary Fig. 46. The whole processes of the experimental setup and the results obtained in the absence (a) and presence (b) of HeLa cells. Thr, thrombin.

Comment 3:

The manuscript would benefit from a more comprehensive review of the literature on allosteric signal transduction systems, conformational cooperativity, and antisense oligonucleotide-based genetic regulation.

Our response:

We have now introduced additional related literatures and discussions on the researches of allosteric signal transduction systems and conformational cooperativity in the revised Introduction section. Meanwhile, more references on the antisense oligonucleotide-based genetic regulation were also added to the revised Introduction section and Results as follows.

Page 3, lines 60-62: “Recent progress in structural and biochemical studies has proven that the concept of the cooperativity has expanded to describing and regulating complex biological processes, including system behaviors and regulatory mechanisms^{24, 25, 30-32}...The examples of riboswitches show that nature RNAs adopt the cooperativity to achieve modulations of gene expression^{37, 38}.”

Page 3, lines 71-79: “Currently, there is considerable interest in constructing artificial allosteric molecular systems, where the molecular information is recognized, transmitted and regulated mainly by conformational signals, instead of chemical signals⁴³⁻⁴⁵. In particular, DNA based allosteric systems have been well developed to construct nanoprobe, biosensors and nanoswitches for applications of controlled drug-release, point-of-care diagnostics and in vivo imaging⁴⁶⁻⁴⁹. In addition, complex artificial molecular systems, combining ligand and oligonucleotide interactions together, have been established to perform a signal transduction function via the allosteric controls, e.g., antibody induced DNA receptors, aptamer binding platforms, nucleic acid nanoswitches⁵⁰⁻⁵⁸.”

Page 7, lines 199-203: “Recently, ASOs have been widely developed as a therapeutic genetic engineering strategy for many diseases⁶⁰⁻⁶². With a growing number of ASOs based therapeutics, many advanced optimizations have been made to improve the delivery efficiency and the target engagement^{59, 61}. These improved ASOs usually adopted the strategies of specific backbone modifications thus conferring enhanced

properties⁶². Meanwhile,..."

Comment 4:

The conclusion section could be strengthened by summarizing the main findings of the study and their potential implications for future research in the field.

Our response:

Based on the reviewer's suggestions, we have summarized the main findings of this research as well as their promising implications for future study in the Discussion section of the revised manuscript. In particular, to better illustrate future applications, an additional picture on the potential applications of CAST system has also been provided as revised Supplementary Fig. 45.

Page 9, lines 266-268: "There are four main findings in our current study, including flexible orthogonal design, precise allosteric regulation, multi-signal conformation cooperativity and logic control gene expression."

Page 10, lines 289-294 : "Therefore, the CAST platform has promising potential applications in the development of future molecular signaling systems to control gene regulation, biosensing, biocomputing, and the treatment of specific diseases. For example, the ligand-bound CAST antisense oligonucleotide technology can be established to regulate target genes for advanced intelligent diagnosis and therapy, in responding to the protein triggers in cellular and complex environmental conditions."

Page 10, lines 307-309 : "The CAST platform represents an avenue to develop future molecular systems and nanomachines for potential applications of variety of biological signal detection and regulation in the related fields."

Revised Supplementary Fig. 45. Potential application scenarios of the CAST system.

Comment 5:

The authors should provide more details on the limitations of the proposed approach, including potential sources of variability and the extent to which the system can be generalized to other applications.

Our response:

We thank the reviewer for the valuable comments. We have now more comprehensively discussed the limitations of our research methods in the revised Discussion section like below.

Page 10, lines 294-302: “On the other hand, the reported systems still have several potential limitations. For example, the operations of CAST process must depend on the specific recognition sites, which makes it difficult to work with arbitrary protein inputs, especially for the proteins without aptamer sequence. In addition, it is challenging to establish large scale and hierarchical signal transduction cascades, due to the difficulties in the designs of complex ligand binding interactions and the controls of unavoidable system leakage. Moreover, as the signal triggering may occur outside or inside cellular

contexts, more stable and delicate CAST receptors will be required to adapt the complex cellular environments. Thus, it is needed to generate a universal and practical CAST platform by overcoming these limitations in the future.”

Comment 6:

It would be useful to include a comparison of the CAST system with other existing methods for regulating gene expression or signal transduction, highlighting the unique advantages and limitations of the CAST system.

Our response:

We thank the reviewer for these valuable comments. The additional comparison and discussion were provided in the Discussion section of the revised manuscript. We also additionally added two supplementary Tables 1 and 2 to in detail compare the CAST and the other existing methods for regulating signal transduction and gene expression in the revised supplementary information.

Page 10, lines 286-294: “Since our work presents a unique cooperative conformation approach, it provides an alternative allosteric signal regulation tool that the signal transductions and gene expressions can be precisely regulated by simply varying the structures of the DNA constructs, instead of the traditional concentration dependent methods. Therefore, the CAST platform has promising potential applications in the development of future molecular signaling systems to control gene regulation, biosensing, biocomputing, and the treatment of specific diseases. For example, the ligand-bound CAST antisense oligonucleotide technology can be established to regulate target genes for advanced intelligent diagnosis and therapy, in response to the protein triggers in cellular and complex microenvironmental conditions.”

Table 1. Comparisons of the CAST and other molecule signal transduction methods						
Author	Triggers	Output	Trigger numbers	Applications	Regulation mode	References
Y. Liang et al.	Thrombin, Streptavidin	Oligonucleotide	1 or 2	ASO Gene regulation	Conformation cooperation, concentration	This work.
Q.L. Zhang et al.	ATP, Thrombin	Oligonucleotide	1	Nucleic acid computation	Concentration, conformation	S1
W. Engelen et al.	Antibody	Oligonucleotide	2	Logic circuit	Concentration	S2
C. Zhang et al.	Oligonucleotide	Oligonucleotide	1	Logic circuit, nanomachines	Conformation cooperation, concentration	S3
S. Ranallo et al.	Antibody	Oligonucleotide	2	Modular DNA-based nanomachine	Concentration	S4
S. Bracaglia et al.	Antibody	Transcribed RNA	2	Cell-free biosensor	Concentration, conformation	S5
P. Li et al.	DNA	ATP	1	Transmembrane Transport	Concentration, light-controlled	S6
L.A.P. Thompson et al.	DNA	ATP	1	Aptamer switches	Concentration, PH	S7

Table 2. Comparisons of the CAST and the current ASO gene regulation methods					
Author	Modification and carrier tape	Regulation mode	Applications	Target	References
Y. Liang et al.	Ligand targeting	Conformation cooperation, concentration, logically control	Gene regulation, Cell apoptosis	GFP, PLK1	This work.
C. Xue et al.	Nanoparticles	Concentration	Cell apoptosis	PLK1	S8
X.H. Wu et al.	Liposome carrying	Conformation, concentration	Gene Therapy	p53	S9
G. A. O. Cremers et al.	DNA nanostructure	Concentration, DNA origami form	Cell surface receptor binding	PD1, EGFR, HER2	S10
B. Cai et al.	Chemical modification	Concentration	Selection of DNA-encoded libraries to protein targets within and on living cells	Halotag-CBX7-ChD, SNAPtag-DOR	S11
Q. Jiang et al.	DNA Origami	Concentration	Cell apoptosis	MCF-7 cell	S12
Y. Zhang et al.	Nanoparticles	Concentration	Tumor therapy	miRNA-21	S13
Z.H. Di et al.	Peptide nucleic acid	Concentration	Tumor therapy	caspase-3	S14
D Hong et al.	Free uptake	Concentration	Tumor therapy	STAT3 RNA	S15

Reviewer 2

This work described a binding-induced strand displacement strategy to achieve signal transduction between proteins and oligonucleotides. Based on this strategy, the authors demonstrated multiple and logic operations between different proteins (thrombin and streptavidin), and further tried to use this system to regulate cellular gene expressions and tumor cell proliferation. The concept of this design seems solid, which may be possible to provide a new protein-oligonucleotide transduction platform. However, I do have some important concerns regarding the generality

Comment 1:

Generality of this design: The strand displacement described in this paper seems to rely on the protein recognition from at least two different binding sites. As investigated in this work, the streptavidin (can bind with four biotin molecules) and the thrombin (can bind with two different aptamers) worked well in this system. However, for the protein with one binding site or with only one available aptamer, would this design be still applicable? The authors tried to examine the performance on transduction of PDGF-BB with only one available aptamer (Supplementary Fig. 14), but these results are not clear. I suggest that comprehensive investigations on proteins with a single binding site or with a single binding aptamer should be performed to check the universality of this design on protein-oligonucleotide transduction in general.

Our response:

We thank the reviewer for these constructive suggestions on our work. Based on the reviewer's suggestion, we have now performed the comprehensive investigations on whether the single binding site can trigger the CAST receptor. With the detailed experiment designs as described below, our results indicate that the design of single-trigger-site CAST is rather difficult due to the complexity in structural changes and more easily influenced by the specific aptamer sequences, when compared with that in multiple-trigger-site CAST. We have made the comprehensive investigations on the CAST regulations by varying the protein concentrations and loop lengths. The detailed experimental methods and results (revised Supplementary Figures 39-44), and the corresponding discussions have been added into the revised manuscript

(Supplementary information section S12). We strongly believe that the addition of these data will further highlight the advantage of our current CAST regulation system and thus improves the quality of manuscript.

Revised Supplementary Fig. 40. Single-trigger-site CAST operations triggered by thrombin. **a,b,c**, Schematic and design illustration of single-trigger-site CAST triggered by thrombin. **d,e**, PAGE gel analysis (**d**) and fluorescence output (**e**) of single-trigger-site CAST. **d**: Lane 1: D; Lane 2: CE-29apt; Lane 3: CE-29apt + Thr; Lane 4: CDE-29apt; Lane 5: CDE-29apt + Thr. **d,e**: [DNA complex] = 0.6 μ M, [Thr] = 1.5 μ M. **f,g**, PAGE gel analysis (**f**) and fluorescence output (**g**) of single-trigger-site CAST varying thrombin concentrations. **f**: [DNA complex] = 0.6 μ M, [Thr] = 0, 0.15 μ M, 0.3

μM , 0.6 μM , 0.9 μM , 1.2 μM , 1.5 μM and 1.8 μM . **g:** [DNA complex] = 0.6 μM , [Thr] = 0, 0.3 μM , 0.6 μM , 0.9 μM , 1.2 μM , and 1.5 μM .

To investigate whether the single binding site can trigger the CAST receptor, we constructed two kinds of single-trigger-site CAST receptors that were designed to be triggered by proteins thrombin and PDGF-BB, respectively. We first designed the single-trigger-site CAST receptor using only one 29 nt aptamer to interact with thrombin protein (Supplementary Fig. 40 a to c). Specifically, the receptor was consisted of three DNA strands as C, D and E-29apt. It should be noted that the DNA E-29apt is designed with 29 nt aptamer sequence in the middle as the single binding site so that the thrombin can bind the 29 nt aptamer to generate a significant conformational changes of the DNA receptor, thus inducing the close proximity of the two arms of hairpin DNA. Therefore, such conformational changes can trigger the CAST effect to release DNA D from the DNA receptor. In the gel results, it is clear to see that a target gel band representing the released DNA D can be observed in lane 5 (Supplementary Fig. 40d). Additionally, the fluorescent results also demonstrated the significant positive signal when triggered by thrombin. Finally, a single-trigger-site CAST experiment was implemented using gradient thrombin concentrations. In the gel results, a gradual increase of gel band intensities can be observed from lanes 2 to 9, when the thrombin concentration was at 0.15 μM , 0.3 μM , 0.6 μM , 0.9 μM , 1.2 μM , 1.5 μM and 1.8 μM (Supplementary Fig. 40 f). Similarly, the gradual increases of fluorescent signals also can be found in Supplementary Fig. 40g. Overall, our results demonstrated the well performances of the single-trigger-site CAST receptor triggered by thrombin.

Next, we test the conformational regulations of single-trigger-site CAST module by introducing loop regulator T3 with the lengths varying from 3 to 30 nt (Supplementary Fig. 41 a and b). Fluorescent assay also demonstrated the down-regulation effects, where the gradual decreasing fluorescent signals were generated with the loop lengths increasing from 3 to 30 nt (Supplementary Fig. 41 c and d). Meanwhile, in the gel results, the gradual decreasing band densities can be observed from lanes 7 to 11, with

the increase of the loop lengths (Supplementary Fig. 41 e and f). Therefore, the experimental results demonstrated that the precise conformational regulations also can be implemented in the single-trigger-site CAST module.

Revised Supplementary Fig. 41. The length variations of T3 to regulate the single-trigger-site CAST ($T_1 = 3$ nt, $T_2 = 12$ nt, $H = 12$ nt). **a,b**, Illustrations (a) and designs (b) of the single-trigger-site CAST triggered by thrombin with different T3 lengths. **c**, Fluorescence results of the single-trigger-site CAST triggered by thrombin with different T3 lengths of 3, 8, 13, 20 and 30 nt for complex-T3 (1), (2), (3), (4), (5). $[\text{complex-T3}] = 0.6 \mu\text{M}$ and $[\text{Thr}] = 1.5 \mu\text{M}$. **d**, Quantification of the final fluorescence values of the single-trigger-site CAST. **e,f**, Native PAGE (12% acrylamide) results (**e**) and statistical analysis (**f**) using ImageJ **f**, of single-trigger-site CAST. Lane 1: D, Lane 2: complex-T3 (1), Lane 3: complex-T3 (2), Lane 4: complex-T3 (3), Lane 5: complex-T3 (4), Lane 6: complex-T3 (5), Lane 7: complex-T3 (1) + Thr, Lane 8: complex-T3 (2) + Thr, Lane 9: complex-T3 (3) + Thr, Lane 10: complex-T3 (4) + Thr, Lane 11: complex-T3 (5) + Thr. $[\text{complex-T3}] = 0.6 \mu\text{M}$ and $[\text{Thr}] = 1.5 \mu\text{M}$.

On the other hand, we also tried to construct another single-trigger-site CAST module triggered by protein PDGF-BB (Supplementary Fig. 43 a to c). At first, we used two kinds of PDGF-BB aptamers PDGF-35apt and PDGF-43apt to serve as the single-trigger-site to initiate the CAST regulations. However, the results indicated that the DNA D was not able to be encapsulated in the initial structure of CAST receptor with aptamers PDGF-35apt. In the gel results in Figure R5d, it is clear to see that a gel band of DNA D was left in lane 4 without hybridizing with the DNA complex. The possible reason may be that the secondary structures of the PDGF-BB aptamer spontaneously aggregate to form a complex self-hybridized structures, thus bringing the two ends of DNA C into a close proximity and making it difficult for DNA D to hybridize with the hairpin DNA C (Supplementary Fig. 43 e and f).

Finally, we tried to construct the structure of CAST receptor with aptamers PDGF-43apt (Supplementary Fig. 43 g to i). Moreover, we tried to improve the PDGF-BB triggered single-site CAST receptor by optimizing the DNA sequences with $H = 11$ nt and the hybridization lengths of C/D as 18 nt (Supplementary Fig. 44). However, the experimental results were still not satisfied. These experimental results indicated the constructions of the single-trigger-site CAST are different from the that of two-trigger-site CAST presented in original the manuscript. The reasons may lie in: 1) The separated design of aptamer trigger sites in the two-trigger-site CAST receptor is an important factor, which avoids the secondary structure to cause the initial aggregations; 2) The single-trigger-site CAST design itself is more susceptible to the influences of the secondary structure induced by the complex aptamer sequences. In addition, it is difficult to overcome even by carefully manual designs. Therefore, the universality of the single-trigger-site CAST is not as general as that of two-trigger-site CAST.

Revised Supplementary Fig. 43. Single-trigger-site CAST operations triggered by PDGF-BB. **a,b,c**, Schematic and design illustration of single-trigger-site CAST operations triggered by PDGF-BB. **d**, PAGE gel analysis of the single-trigger-site CAST operations triggered by PDGF-BB (aptamer PDGF-35apt). Lane 1: D; Lane 2: CE-35apt; Lane 3: CE-35apt + PDGF-BB; Lane 4: CDE-35apt; Lane 5: CDE-35apt + PDGF-BB. [DNA complex] = 0.6 μ M, [PDGF-BB] = 1.5 μ M. **e**, NUPACK simulation results of the assembly of strands C, and E-35apt (at 37°C and 1 μ M concentration). **f**, NUPACK simulation results of the assembly of strand E-35apt (at 37°C and 1 μ M concentration). **g**, PAGE gel analysis of the single-trigger-site CAST operations triggered by PDGF-BB (aptamer PDGF-43apt). Lane 1: D; Lane 2: CE-43apt; Lane 3: CE-43apt + PDGF-BB; Lane 4: CDE-43apt; Lane 5: CDE-43apt + PDGF-BB. **d,e**: [DNA complex] = 0.6 μ M, [PDGF-BB] = 1.5 μ M. **h**, NUPACK simulation results of

the assembly of strands C, and E-43apt (at 37°C and 1 μM concentration). **i**, NUPACK simulation results of the assembly of strand E-43apt (at 37°C and 1 μM concentration).

Revised Supplementary Fig. 44. Structural optimization of single-trigger-site CAST operations triggered by PDGF-BB. **a,b**, Structural optimization of single-trigger-site CAST operations triggered by PDGF-BB. **c**, PAGE gel analysis of the single-trigger-site CAST operations triggered by PDGF-BB. Lane 1: D; Lane 2: C-1DE-35apt; Lane 3: C-1DE-35apt + PDGF-BB; Lane 4: C-1DE-43apt; Lane 5: C-1DE-43apt + PDGF-BB. [DNA complex] = 0.6 μM, [PDGF-BB] = 1.5 μM.

Comment 2:

Experiments in cells: The procedures on applications of this system in cellular manipulation are ambiguous. In Line 201-204, it seems that the transduction system was firstly incubated with the thrombin in test tubes to release the output strand before transfection into the cells. If so, all these cellular experiments (Fig. 4 and 5) would be meaningless in terms of gene regulation, as the transduction processes occurred outside cellular contexts instead of generation of the functional oligonucleotides in situ. The

authors must clearly state how they performed these cell experiments. If the transduction could not be demonstrated to occur inside cells, the significance of these results would not be as important as stated in the manuscript.

Our response:

We thank the reviewer for raising this important concern. In our present study, the CAST transduction procedures were indeed triggered by input proteins outside the cellular contexts, and then released the target ASO oligonucleotides entering into the cell to regulate the expressions of the target genes. Actually, the main purpose of our study is to develop CAST transduction systems that can release target ASO or therapeutic drugs only in response to tumor microenvironment-specific extracellular stimuli (e.g. to overexpress proteins or metabolites), so as to achieve targeted gene regulations or drug delivery and accumulation in tumor tissues (Decai Zhao et al. *Adv. Funct. Mater.*, 2023; M. Shahriari et al. *J. Controlled Release*, 2019). Therefore, the CAST systems are designed to be triggered outside cellular contexts in response to specific environmental stimuli. Our further work will focus on investigating whether the thrombin-triggered CAST system works well around tumor sites in tumor-bearing mouse models, because thrombin protein is specifically overexpressed in the tumor microenvironment (Suping Li. et al. *Nat. Biotechnol.*, 2018; Bing Zhao et al. *Signal Transduct. Tar.*, 2020). The future related study will be presented in another manuscript and is beyond the scope of our present work. To make this point more clear, we have now properly added the corresponding discussion in the revised manuscript.

In fact, the concern raised by the reviewer is a crucial aspect in the fields of DNA based bioengineering. Recently, functional nucleic acids associated with DNA nanotechnology have been developed to construct dynamic nanodevices that recognize specific signals, and in turn produce corresponding responses for realizing complex bioengineering (Bertucci, A. et al. *Angew. Chem. Int. Ed.*, 2020; Yan, X. et al. *Angew. Chem. Int. Ed.*, 2015). It has been reported to make significant progresses that there have numerous applications in disease diagnosis, detection, cancer therapy and genetic engineering (Dong, H. L. et al. *ACS Appl. Bio Mater.*, 2020). Based on the signal transduction modes, the DNA nanodevices can be divided into two main categories,

depending on whether locations of the signal triggering sites are intracellular (e.g., in cytoplasm) or extracellular (e.g., on cell surface, in plasma) (Figure R7, Table 3 and 4).

For the former, the process of the DNA receptors recognizing the signals occurs within cells, and thus directly induces the next reactions in cellular contexts in situ. For example, Peng et al. have successfully demonstrated the implementation of a microRNA-activated DNAzyme motor that effectively operates within living cells (Peng, H. Y. et al. *Nat. Commun.*, 2017). This innovative approach enables the recognition of miRNA and subsequent activation of the DNAzyme motor, thereby facilitating amplified imaging of miRNA in situ within cells.

For the latter, the signals trigger the DNA receptors outside the cellular contexts, and the corresponding response induces the following biofunctions at cellular targets via a transmembrane signal transduction. In 2020, Tian et al. have developed a nanorobot using a DNA framework to selectively bind to ligand receptors on target cell surface (Tian, T. R. et al. *Adv. Funct. Mater.*, 2020). Through allosteric activation outside the cells, the nanorobot releases melittin that effectively kills neighboring tumor cells and realizes targeted therapy for tumor cells.

Therefore, these researches have clearly shown that the specific biofunctions can be well performed or engineered by operating the DNA nanodevices in both extracellular and intracellular ways. In other words, the working effects of the DNA nanodevices are independent on whether the signal triggering occurs outside or inside cellular contexts. And adopting what kinds of signal transduction modes mainly depends on the specific purposes and the requirements of the DNA implementations. In general, although the CAST system exhibits initial promising effects and potential applications in bioengineering fields, more future researches are still needed to deeply investigate the possibility to trigger the CAST mechanism in more complex and precise manners.

Considering the importance of the questions raised by the reviewer, we made the corresponding revisions in the following aspects.

(1) Making clear statements of the signal transduction modes in the revised experimental processes. The descriptions are added in the revised methods section.

Page 15, lines 480-486:

“Cell experiment procedures

The DNA transducer was annealed at a concentration of 5 μ M. After annealing, the triggering protein was added and incubated at room temperature for 2h. Then the DNA samples were co-incubated with Hela cells (with the final DNA concentration as 500 nM), and the cell incubation time specifically changed in different experiments. In the cell uptaking experiments, the cell incubation time was 2 hours. In the gene regulation experiment, the cell incubation time was about 48-72 hours.”

(2) Adding detailed illustrations of cell experiment procedure in the revised supplementary information to help readers understand the CAST signal transduction modes (Revised Supplementary Fig. 46).

The related descriptions were made in the revised supplementary materials as following.

For the in vitro experiments, short DNA strands with equal concentrations were mixed and annealed at 65°C (when modified DNA was used in the experiments, otherwise annealing was performed at 95°C). The protein with corresponding concentrations was then added and incubated for 2 hours before performing PAGE and fluorescence experiments. For the cellular experiments, the annealed DNA transducers were treated with protein triggers (e.g., thrombin and streptavidin) to release single-stranded oligonucleotides outputs. Subsequently, the DNA/protein samples from each experimental group were co-incubated with Hela cells by incubating certain time periods. The experimental results were monitored by multiple instruments including flow cytometry, confocal microscopy, and western blot analysis, et al.

Revised Supplementary Fig. 46. The whole processes of the experimental setup and the results obtained in the absence (a) and presence (b) of HeLa cells. Thr, thrombin.

(3) Adding additional Figures and Tables in the supplementary information for the detail comparisons of the two kinds of DNA devices with intracellular and extracellular working mechanisms (Revised Supplementary Fig. 47, Table 3 and 4).

DNA devices triggered in extracellular context

DNA devices triggered in intracellular context

Revised Supplementary Fig. 47. DNA devices with intracellular and extracellular working mechanisms.

Table 3. DNA systems for the extracellular triggering bioengineers.					
Author	Name	Applications	Target	Trigger Pathway	References
S.M. Traynor et al.	Bio-barcode assay	Cancer detection	Prostate specific antigen (PSA)	Human plasma/extracellular	S15
B. Koos et al.	proxHCR	Protein Interactions and posttranslational modifications	E-cadherin and b-catenin in DLD1 cells	Cell surface /extracellular	S16
L. Li et al.	Structure-switchable aptamer (SW-Apt)	Modulating aptamer specificity	PTK-7	Cell surface /extracellular	S17
S.P. Li et al.	DNA nanorobot	Cancer therapeutic	Nucleolin	Cell surface /extracellular	S18
Y.Y. Sun et al.	DNA- origami-based pMHC multimers	Antigen-specific CD8+ T cell detection	T cell	Cell surface /extracellular	S19
T. Shibata et al.	Protein-driven RNA nanostructured devices	Regulate mammalian cell fate	L7Ae	Extracellular	S20
Y.J. Wang et al.	Xeno-nucleic-acid-modified classic DNAzyme	Silences gene expression	RNA substrate	Extracellular	S21
M.S. Xiao et al.	DNA reaction circuits	Programming multiple cell-cell interactions	Multiple cell	Extracellular	S22
J. Li et al.	3D amphiphilic pyramidal DNA	Cellular interactions	Multiple cell	Extracellular	S23

Table 4. DNA based gene regulations with outside cellular triggering.					
Author	Name	Applications	Target	Trigger Pathway	References
K. Jiao et al.	Topologically Ordered DNA	RNA transcription	T7 promoter	Extracellular	S25
Y.J. Wang et al.	Xeno-nucleic-acid-modified classic DNAzyme	Silences gene expression	RNA substrate	Extracellular and intracellular	S22
T. Shibata et al.	Protein-driven RNA nanostructured devices	Regulate mammalian cell fate	L7Ae	Extracellular	S21
D. Han et al.	ssDNA probe (Apt-S-T)	Cell isolation	Sgc8c-S-T1, TCO1-S-T2, Sgc4f-S-T3	Extracellular	S26
K. Zagorovsky et al.	MNAzyme	Disease diagnosis	RNA substrate	Extracellular	S27
S. Angerani et al.	Ligands functionalized with peptide nucleic acids	Responsive membrane dimer protein	Carbonic anhydrases	Cell surface /extracellular	S28
Ishaqat A et al.	CpG ODNs	Immunostimulation	TLR9	Extracellular	S29

(4) Introducing more discussions on the intracellular DNA regulations in future DNA regulated bioengineering. The revisions were made in the third paragraphs in the discussion section.

Page 10, lines 299-302: “Moreover, as the signal triggering may occur outside or inside cellular contexts, more stable and delicate CAST receptors will be required to adapt the complex cellular environments. Thus, there is a strong urge to develop an universal and practical CAST platform by overcoming these limitations in the future.”

Comment 3:

A related report (J. Am. Chem. Soc. 2013, 135, 7, 2443–2446) should be cited. This previous study on binding-induced strand displacement needs to be mentioned, compared and discussed.

Our response:

We sincerely appreciate these valuable comments. We have now added this report and other more related references on allosteric signal transduction systems and binding-induced strand displacement, and given the corresponding discussion in the Introduction section of the revised manuscript as below.

Page 3, lines 76-79: “**In addition, complex artificial molecular systems, combining ligand and oligonucleotide interactions together, have been established to perform a signal transduction function via the allosteric controls, e.g., antibody induced DNA receptors, aptamer binding platforms, nucleic acid nanoswitches⁵⁰⁻⁵⁸.**”

Comment 4:

Some experimental procedures are not clearly described. For instance, how were cells treated with these oligonucleotides? By transfection? In what concentrations?

Our response:

We thank the reviewer for pointing out these points. We have now provided more detailed descriptions about these experiments in the revised manuscript as below.

Page 15, lines 480-486:

“Cell experiment procedures

The DNA transducer was annealed at a concentration of 5 μ M. After annealing, the triggering protein was added and incubated at room temperature for 2h. Then the DNA samples were co-incubated with Hela cells (with the final DNA concentration as 500 nM), and the cell incubation time specifically changed in different experiments. In the cell uptaking experiments, the cell incubation time was 2 hours. In the gene regulation experiment, the cell incubation time was about 48-72 hours.”

Comment 5:

The naming rule for output oligonucleotides is not defined. For instance, in Fig. 1b, what is the difference between Oligo D, D1 and P? These symbols are not explained in the manuscript.

Our response:

We thank the reviewer for reminding us of these important points. The names of output oligonucleotides have now been defined in detail in the revised legends of Figure 1 and Figure 3. The corresponding descriptions have also been added into the revised text and figure of manuscript.

Page 18, lines 553-555: “...Oligo D, Oligo D_{OR}, Oligo D_{AND} and Oligo D_{CD} are the oligonucleotide outputs of the basic AST and CAST, OR logic operation, AND logic operation and cascading CAST logic operation, respectively.”

Page 20, lines 574-587: **“Fig. 3 Two-input logic operations based on the CAST strategy. a,b**, Schematic illustration of an OR logic gate. **c**, PAGE results of OR logic operation using DNA complex-or (CD_{OR}A1B1). Lane 1: D_{OR}; Lane 2: CA1B1; Lane 3: CA1B1 + Thr; Lane 4: CA1B1 + SA; Lane 5: complex-or; Lane 6: complex-or + Thr; Lane 7: complex-or + SA; Lane 8: complex-or + Thr + SA. **d**, Fluorescence assay of OR logic operation. **e,f**, Schematic illustration of AND logic gate. **g**, PAGE results of

AND logic operation using DNA complex-and (C1B2 + C2A2D_{AND}). Lane 1: D_{AND}; Lane 2: C1B2 + C2A2; Lane 3: C1B2 + C2A2 + Thr; Lane 4: C1B2 + C2A2 + SA; Lane 5: C1B2 + C2A2 + Thr + SA; Lane 6: complex-and; Lane 7: complex-and + Thr; Lane 8: complex-and + SA; Lane 9: complex-and + Thr + SA. **h**, Fluorescence assay of AND logic operation. [DNA strands] = 0.6 μ M, [Thr] = 1.5 μ M, [SA] = 1.8 μ M. **i**, Cascading CAST circuit triggered by thrombin and streptavidin. **j**, The illustrations and **k**, designs of cascading CAST logic operation (using DNA complex (XI), ZS*XY and complex (XII), OD_{Ca}R*S), respectively. S* and D_{Ca} are the upstream and downstream outputs of the cascade circuit, respectively. **l**, Fluorescence results. [complex (XI)] = 0.5 μ M, [complex (XII)] = 0.5 μ M, [Thr] = 1.25 μ M, [SA] = 1.5 μ M.”

Reviewers' Comments:

Reviewer #1:

Remarks to the Author:

The authors have well responded to my concerns.

Reviewer #2:

Remarks to the Author:

The authors have addressed some of my concerns in their response, but how they present the revised manuscript is not suitable for the readership. So many important designs and experiments are only placed in the supplementary material without even a description in the main text. This is not the proper way to present an intact study. I suggest some important revisions for the main text as followings.

1. Supplementary Fig. 46 should be moved into the main text to give a clear message regarding how the experiments performed. I suggest placing Supplementary Fig. 46a within Fig. 1 and Supplementary Fig. 46a within Fig. 4 in the main text.

2. The authors introduced a different but related design to show that their system could also work with one binding aptamer in the supplementary material. These results are important and should be comprehensively described in the main text. However, the authors did not even mention this design in the main text. I strongly suggest a new main figure for description of this design and the related experimental results to show the generality of their system.

3. As the authors mentioned in their rebuttal, their transduction process for ASO experiments occurred in the test tubes rather than inside cells. The authors argued that the extracellular environment could be also significant for application of these DNA nanomachines, but their experimental setup in test tubes is far away from this context. This is an important limitation of their current study that must be mentioned and discussed in the main text.

Reviewer 2

The authors have addressed some of my concerns in their response, but how they present the revised manuscript is not suitable for the readership. So many important designs and experiments are only placed in the supplementary material without even a description in the main text. This is not the proper way to present an intact study. I suggest some important revisions for the main text as followings.

Comment 1:

Supplementary Fig. 46 should be moved into the main text to give a clear message regarding how the experiments performed. I suggest placing Supplementary Fig. 46a within Fig. 1 and Supplementary Fig. 46b within Fig. 4 in the main text.

Our response:

We thank the reviewer for the constructive suggestion that will help readers to better understand the experimental processes of our work. Based on the reviewer's suggestion, we have added the experimental process to the main text Fig. 1e and Fig. 5b. The revised Fig.1 and Fig.5 are shown below.

Fig. 1 An allosteric protein-oligonucleotide signal transduction system. a,b, Schematic illustrations of allosteric protein-oligonucleotide signal transduction mechanisms (a), cooperative allosteric signal transduction networks (b), respectively. Oligo D, Oligo D_{OR}, Oligo D_{AND} and Oligo D_{CD} are the oligonucleotide outputs of the basic AST and CAST, OR logic operation, AND logic operation and cascading CAST logic operation, respectively. **c**, Design of a basic allosteric signal transduction system, respectively. 15Apt and 29Apt: Two aptamers of thrombin; FAM: fluorophore; BHQ1: quencher. **d**, PAGE gel analysis of the basic allosteric transduction system. Lane 1: D; Lane 2: CAB; Lane 3: CAB + Thr; Lane 4: CDAB; Lane 5: CDAB + Thr. [DNA complex] = 0.6 μM, [Thr] = 1.5 μM. **e**, The whole experimental processes setup in vitro. **f,g**, Fluorescence output of the basic allosteric transduction system triggered by

thrombin (f) and the reactions with varying thrombin concentrations (g), respectively. **f:** [CDAB] = 0.6 μ M, [Thr] = 1.5 μ M. **g:** [CDAB] = 0.6 μ M, [Thr] = 0, 0.3 μ M, 0.6 μ M, 0.9 μ M, 1.2 μ M and 1.5 μ M. **h,** Fluorescence output of the basic allosteric transduction system triggered by streptavidin. [C6*D3*A2B2] = 0.6 μ M, [SA] = 0, 0.6 μ M, 1.2 μ M, 1.8 μ M, 2.4 μ M and 3 μ M.

Fig. 5 Using CAST based ASOs to regulate cellular gene expression. **a**, Schematic illustration of the allosteric regulation of GFP gene expression via thrombin. **b**, The whole experimental processes setup in vivo. **c**, Confocal microscopy imaging results. Scale bars: 200 μ m. **d,e**, Schematic illustration (**d**) and the confocal microscopy images (**e**) of CAST regulation of GFP gene expression using two regulators T1 and T2, respectively. Scale bars: 200 μ m. **f,g**, Relative GFP-positive cells (**f**) and mean fluorescence intensities (**g**) of CAST-regulated GFP gene expression, respectively. **f,g**: Data collected in (**e**) were quantified using ImageJ software and are presented as mean \pm s.d. for $n = 3$ biologically independent experiments. Source data are provided as a Source Data file.. Statistic analysis for (**f**) and (**g**) was performed using two-sided test (* $p \leq 0.05$, ** $p \leq 0.01$, *** $p \leq 0.001$).

Comment 2:

The authors introduced a different but related design to show that their system could also work with one binding aptamer in the supplementary material. These results are important and should be comprehensively described in the main text. However, the authors did not even mention this design in the main text. I strongly suggest a new main figure for description of this design and the related experimental results to show the generality of their system.

Our response:

We thank the reviewer for this important suggestion. After careful consideration, we decided to introduce a new section in the main text to describe the designs and the experimental results of the single-trigger-site CAST operations. The introduced text and figure are shown as following.

“The CAST operations with single-trigger-site

To investigate whether can be triggered by the CAST receptor with the single binding site, we constructed two kinds of CAST receptors that were designed to respond to any one of thrombin or PDGF-BB. We first designed the single-trigger-site CAST receptor using one 29 nt aptamer to interact with protein thrombin (Fig 4a, b, and Supplementary Fig. 29). It should be noted that 29 nt length DNA E-29apt is designed in the middle as

the single binding site. Therefore, binding of thrombin to the 29 nt aptamer sequence can generate a significant conformational changes in the DNA receptor, thus inducing the close proximity of the two arms of hairpin DNA to release DNA D. In the gel results, it is clear to see that a target gel band representing the released DNA D was produced in lane 5 (Fig. 4c and Supplementary Fig.30a-c). Additionally, a significant positive fluorescent signal was produced when triggered by thrombin. The gradual increases of fluorescent signals also can be found with gradient increasing thrombin (Fig. 4d). Overall, our results demonstrated the well performance of the single-trigger-site CAST receptor triggered by thrombin protein.

Next, we also tested the single-trigger-site based down-regulation module by introducing loop regulator T3 with the lengths varying from 3 to 30 nt (Fig. 4 e and f). Fluorescent assay was implemented, and the responding results showed the down-regulations where the gradual decreasing fluorescent signal densities were generated with the loop lengths increasing (Fig. 4g and Supplementary Fig. 30d). Meanwhile, in the gel results, the gradual decreasing band densities of released DNA D can be observed with the increase of loop lengths (Supplementary Fig. 30e and f). The experimental results demonstrated that the precise regulations also can be implemented in the single-trigger-site CAST module. We also tried to construct another single-trigger-site initiated CAST receptor by using protein PDGF-BB (Supplementary Fig. 31-33).”

Fig. 4 Single-trigger-site CAST operations triggered by thrombin. **a,b**, Schematic and design illustration of single-trigger-site CAST triggered by thrombin. **c**, PAGE gel analysis of single-trigger-site CAST. Lane 1: D; Lane 2: CE-29apt; Lane 3: CE-29apt + Thr; Lane 4: CDE-29apt; Lane 5: CDE-29apt + Thr. [DNA complex] = 0.6 μ M, [Thr] = 1.5 μ M. **d**, Fluorescence output of single-trigger-site CAST varying thrombin concentrations. **d**: [DNA complex] = 0.6 μ M, [Thr] = 0, 0.3 μ M, 0.6 μ M, 0.9 μ M, 1.2 μ M, and 1.5 μ M. **e,f**, Illustrations (**e**) and designs (**f**) of the single-trigger-site CAST triggered by thrombin with different T3 lengths, respectively. **g**, Fluorescence results of the single-trigger-site CAST triggered by thrombin with different T3 lengths of 3, 8, 13, 20 and 30 nt for complex-T3 (1), (2), (3), (4), (5). [complex-T3] = 0.6 μ M and [Thr] = 1.5 μ M.

Comment 3:

As the authors mentioned in their rebuttal, their transduction process for ASO experiments occurred in the test tubes rather than inside cells. The authors argued that the extracellular environment could be also significant for application of these DNA nanomachines, but their experimental setup in test tubes is far away from this context. This is an important limitation of their current study that must be mentioned and discussed in the main text.

Our response:

We sincerely thank the reviewer for the valuable comments and we realize the importance of the suggestion. Therefore, we added more discussions in the Discussion section of the revised main text to better explain and explore the possible solutions for the problems. The corresponding discussion of the revised manuscript is shown as following.

“Furthermore, our CAST trigger reactions currently occur in the test tubes rather than extracellular environment. Therefore, there are still some issues associated with the performances induced by extracellular or intracellular triggering. Due to the complexities of the extracellular environments, the following factors should be considered including enzymatic degradation, pH sensitivity, and the efficient delivery of the DNA strands into cells^{59, 65}. To overcome these hurdles, some related technologies can be introduced into CAST systems. One possible approach involves using chemical modification groups to protect the DNA receptors and released ASOs (e.g., phosphorylation by nucleic acid strands), to prevent ribozyme degradation^{61, 62}. Additionally, the carrier based methods also can be introduced to improve the ASOs delivery efficiency and avoid the enzymatic degradations, e.g., liposomal, nanoparticles, polymers^{61, 63, 65}. In general, our CAST method can be improved in many aspects in future researches to develop a more versatile and practical gene regulation platform.”